# The crystal structure of SUN1-KASH6 reveals an asymmetric LINC complex architecture compatible with nuclear membrane insertion

Manickam Gurusaran[1], Benedikte S. Erlandsen[1] & Owen R. Davies [1✉]

The LINC complex transmits cytoskeletal forces into the nucleus to control the structure and movement of nuclear contents. It is formed of nuclear SUN and cytoplasmic KASH proteins, which interact within the nuclear lumen, immediately below the outer nuclear membrane. However, the symmetrical location of KASH molecules within SUN-KASH complexes in previous crystal structures has been difficult to reconcile with the steric requirements for insertion of their immediately upstream transmembrane helices into the outer nuclear membrane. Here, we report the crystal structure of the SUN-KASH complex between SUN1 and JAW1/LRMP (KASH6) in an asymmetric 9:6 configuration. This intertwined assembly involves two distinct KASH conformations such that all six KASH molecules emerge on the same molecular surface. Hence, they are ideally positioned for insertion of upstream sequences into the outer nuclear membrane. Thus, we report a SUN-KASH complex architecture that appears to be directly compatible with its biological role.

[1] Wellcome Centre for Cell Biology, Institute of Cell Biology, University of Edinburgh, Michael Swann Building, Max Born Crescent, Edinburgh EH9 3BF, UK. ✉email: owen.davies@ed.ac.uk

The Linker of Nucleoskeleton and Cytoskeleton (LINC) complex crosses both inner and outer nuclear membranes of the nuclear envelope to mechanically transduce cytoskeletal forces to nuclear contents[1–3]. It is formed of nuclear SUN (Sad1 and UNC84 homology) and cytoplasmic KASH (Klarsicht, ANC-1, and Syne homology) proteins, which interact within the nuclear lumen[1–3]. In mammals, there are five SUN proteins (SUN1-5) and multiple isoforms of six KASH proteins (Nesprin-1-4, KASH5 and JAW1/LRMP). These combine to perform the widespread roles of the LINC complex in nuclear structure, shape and positioning[4–6], in addition to specialised roles in functions such as sound perception in the inner ear and chromosome movements in meiosis[7–10]. Thus, the LINC complex is necessary for the maintenance of nuclear structure in normal cellular life, in addition to hearing and fertility. Further, LINC complex mutations in humans have been implicated in laminopathies, including Hutchison-Gilford progeria syndrome and Emery-Dreifuss muscular dystrophy[11,12].

The physical interaction between SUN and KASH proteins occurs immediately below the outer nuclear membrane, within the nuclear lumen (Fig. 1a). This is mediated by the globular SUN domain at the C-terminus of the SUN protein, and the approximately thirty amino-acid KASH domain at the C-terminal end of the KASH protein[13]. The widespread functions of SUN proteins are performed by generally expressed, and partially redundant, SUN1 and SUN2[14,15]. The upstream sequences of SUN proteins form trimers and higher-order structures[16–19], which traverse the nuclear lumen, cross the inner nuclear membrane, and interact with nuclear contents, including the nuclear lamina, chromatin and the telomeric ends of meiotic chromosomes[6,20–23]. In KASH proteins, transmembrane helices (that cross the outer nuclear membrane) immediately precede the SUN interaction, and lead to diverse cytoplasmic N-terminal domains. The spectrin-repeat domains of widely expressed

Nesprin-1 and Nesprin-2 bind to actin[24–26]. Nesprin-3 interacts via plectin with intermediate filaments[27]. Nesprin-4 functions in sound perception in the inner ear by interacting via kinesin with microtubules[7,10]. KASH5 functions in meiotic chromosome movements by acting as a transmembrane dynein activating adapter of microtubules[8,28–30]. JAW1/LRMP (lymphoid restricted membrane protein) is an atypical KASH protein that is localised in the endoplasmic reticulum and outer nuclear membrane, and is specifically expressed in certain cells of the immune system, small intestine, pancreas and taste buds[31–33]. It interacts with SUN1 and microtubules, and is required to maintain nuclear shape and Golgi structure[31,32].

The architecture of the LINC complex underpins its mechanism of force transduction. At the heart of this is the structure of the SUN-KASH complex, which lies immediately below the outer nuclear membrane. Crystal structures of SUN-KASH complexes have demonstrated that globular SUN domains form 'three-leaf clover'-like structures that emanate from upstream trimeric coiled-coils[13,34,35]. KASH domains are bound at the interface between adjacent protomers, with their C-termini inserted into a KASH-binding pocket and their N-termini interacting with KASH-lids, which are flexible beta hairpin loops within the SUN domain[13,34]. On the basis of early structures using Nesprin-1/2 sequences, it was assumed that SUN-KASH complexes have a 3:3 stoichiometry[13,34]. This agreed with the trimeric nature of the luminal regions of SUN proteins, and provided a simple model, in which a linear LINC complex traverses the nuclear envelope and nuclear luminal to transmit forces from the cytoskeleton[36].

In recent crystallographic studies, it was found that non-canonical KASH domains from Nesprin-4 and KASH5 form 6:6 complexes with both SUN1 and SUN2 proteins, in which two 3:3 SUN-KASH trimers interact head-to-head[37,38]. These head-to-head interactions are mediated by zinc-coordination between opposing KASH domains of the Nesprin-4 structure, and through

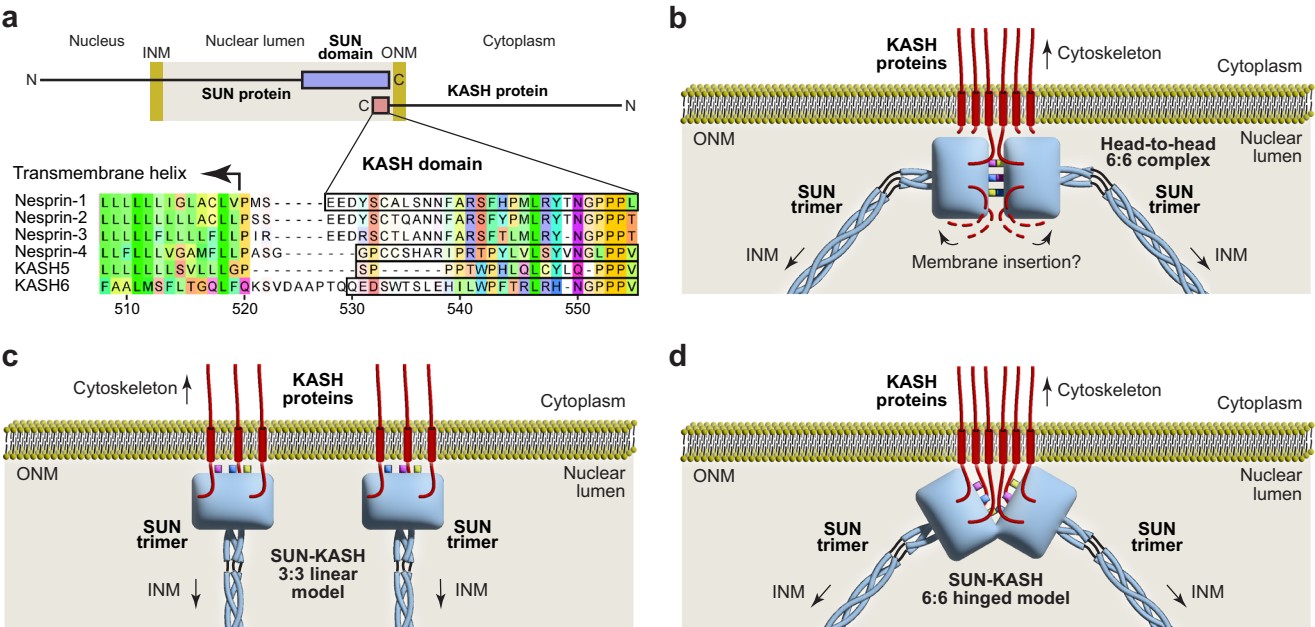

**Fig. 1 Models of the LINC complex. a** Schematic of the LINC complex topology (top), and structure-based alignment of KASH domains from human KASH proteins in which sequences observed in their crystal structures are highlighted (bottom). The KASH domain of JAW1/LRMP is referred to as KASH6 throughout this manuscript. Inner and outer nuclear membranes, INM and ONM, respectively. **b** Schematic highlighting the steric challenge of inserting transmembrane helices of all six KASH molecules into the outer nuclear membrane from the symmetric 6:6 hetero-oligomer observed in crystal structures of the SUN-KASH complex. The KASH-lids of each SUN trimer are coloured blue, yellow and pink. **c, d** Proposed models of LINC complex structure in vivo. **c** Linear model in which SUN-KASH 6:6 complexes are disrupted into 3:3 structures that are oriented vertically in the nuclear lumen[13,34]. **d** Hinged model in which SUN-KASH 6:6 complexes are angled to facilitate insertion of their bound KASH molecules into the ONM[37].

extensive protein-protein interactions between opposing KASH domains of the KASH5 structure[37]. Importantly, it was demonstrated that these 6:6 'head-to-head' complexes represent the solution states of SUN-KASH complexes with Nesprin-4 and KASH5[37]. In the original SUN-KASH structures with Nesprin-1/2, similar head-to-head interactions between 3:3 complexes were present in the crystal lattice, but were assumed to be crystal contacts[13,34]. However, SUN1-KASH1 was found to exist solely as a 6:6 complex in solution, and a point mutation targeting the head-to-head interaction, but not affecting constituent 3:3 structures, completely abrogated complex formation in vitro[37]. Hence, existing biochemical data indicate that the SUN-KASH complex exists in a 6:6 head-to-head configuration that is essential for its stability.

Whilst the SUN-KASH 6:6 head-to-head complex is consistent with existing biochemical data, little sequence is present between KASH domains and their upstream transmembrane helices. Hence, it is difficult to reconcile how it is sterically possible for all six KASH proteins of a symmetric SUN-KASH 6:6 complex to insert into the outer nuclear membrane (Fig. 1b)[3]. There are currently two hypotheses for how this may occur in vivo. Firstly, that SUN-KASH is disrupted into 3:3 complexes that are oriented vertically such that bound KASH proteins are suitably positioned for insertion of their transmembrane helices into the outer nuclear membrane (Fig. 1c)[13,34]. This is a simple and appealing model, but contradicts existing biochemical data as the SUN-KASH complex has been shown to dissociate in solution upon disruption of the 6:6 interface[37]. Further, this model suggests that cytoplasmic KASH proteins must be either monomers or trimers, whereas KASH5 has been shown to be dimeric[29,30]. The second hypothesis is that SUN-KASH 6:6 complexes undergo a hinging motion from their head-to-head configuration, such that all six bound KASH proteins are suitably positioned for membrane insertion (Fig. 1d)[37]. This model is supported by computational and biophysical data indicating that such motions are possible and occur in solution[37], and can explain cytoplasmic KASH proteins forming monomers, dimers or trimers. However, we lack compelling structural evidence in favour of either model.

Here, we report the crystal structure of the SUN-KASH complex between SUN1 and the non-canonical KASH domain of JAW1/LRMP (herein referred to as KASH6; Fig. 1a). This reveals an asymmetric 9:6 structure in which three SUN1 trimers are arranged in a triangular configuration, with two distinct KASH peptides bound to each SUN1 trimer. The KASH peptides adopt conformations that are distinct from previous structures, and are arranged such that the N-termini of all six peptides are positioned on the top surface of the molecule. This configuration appears to be compatible with the insertion of upstream KASH sequences into the outer nuclear membrane. We find that the same structure is formed in solution upon transient treatment of the 6:6 complex in mildly acidic conditions. Hence, we report a SUN-KASH complex crystal structure that appears to be directly compatible with the steric requirements for insertion of immediately upstream KASH sequences into the outer nuclear membrane.

## Results

**SUN1-KASH6 forms an asymmetric 'trimer-of-trimers' structure.** In previous SUN-KASH crystal structures, a conserved mechanism of KASH-binding has been observed in which the C-terminus of the KASH domain is inserted into a binding pocket at the interface between adjacent SUN protomers[13,34,37,38]. However, divergent sequences of the upstream KASH domain interact differently with the associated KASH-lid of the SUN domain, and have different mechanisms of assembly[13,34,37,38]. Indeed, the head-to-head interface between SUN trimers is mediated solely by

KASH-lids in Nesprin-1/2 structures, by zinc-coordination between opposing KASH domains in KASH4 structures, and by intertwined interactions between opposing KASH domains and KASH-lids of KASH5 structures[13,34,37,38].

The recently identified KASH domain of JAW1/LRMP (herein referred to as KASH6) diverges from the canonical sequence of Nesprin-1-3 and the non-canonical sequences of KASH4 and KASH5 (Fig. 1a). Hence, we wondered whether it may adopt a distinct conformation from those previously observed. Thus, we crystallised the SUN1-KASH6 complex and solved its X-ray crystal structures from several crystals at resolutions between 1.7 and 2.3 Å (Table 1 and Supplementary Fig. 1). In contrast with previous SUN-KASH 6:6 structures, the SUN1-KASH6 crystal structure contains nine SUN domains, such that head-to-head interfaces of three SUN1 trimers interact together around a three-fold symmetry axis in a 'trimer-of-trimers' configuration (Fig. 2a, b). Hence, the SUN1-KASH6 crystal structure defines a distinct LINC complex architecture.

Within the trimer-of-trimers structure, SUN1 trimers are positioned with two of their KASH-binding pockets on the sides of the molecule, located at the inter-trimer interface with

**Table 1 Data collection, phasing and refinement statistics.**

|  | SUN1-KASH6 (1.7 Å) 9:9 complex | SUN1-KASH6 (2.3 Å) 9:6 complex | SUN1-KASH6 (2.0 Å) 9:6 complex |
|---|---|---|---|
| **PDB accession** | 8B46 | 7Z8Y | 8B5X |
| **Data collection** |  |  |  |
| Space group | P6₃ | P6₃ | P6₃ |
| Cell dimensions |  |  |  |
| *a, b, c* (Å) | 134.09, 134.09, 106.59 | 133.76, 133.76, 106.71 | 133.92, 133.92, 106.58 |
| α, β, γ (°) | 90, 90, 120 | 90, 90, 120 | 90, 90, 120 |
| Resolution (Å) | 116.13–1.67 (1.89–1.67)ᵃ | 78.49–2.29 (2.52–2.29)ᵃ | 115.98–1.98 (2.22–1.98)ᵃ |
| Ellipsoidal diffraction limits (Å) with principal axes | 2.21 (1, 0, 0) 0.894 aᵃ - 0.447 bᵃ 2.21 (0, 1, 0) bᵃ 1.63 (0, 0, 1) cᵃ | 2.79 (1, 0, 0) 0.894 aᵃ - 0.447 bᵃ 2.79 (0, 1, 0) bᵃ 2.22 (0, 0, 1) cᵃ | 2.60 (1, 0,0) 0.894 aᵃ - 0.447 bᵃ 2.60 (0, 1, 0) bᵃ 1.92 (0, 0, 1) cᵃ |
| $R_{meas}$ | 0.140 (1.987) | 0.333 (2.253) | 0.221 (2.150) |
| $R_{pim}$ | 0.043 (0.622) | 0.074 (0.499) | 0.048 (0.458) |
| $I / \sigma(I)$ | 17.4 (1.9) | 10.6 (1.8) | 12.5 (1.8) |
| $CC_{1/2}$ | 0.999 (0.676) | 0.995 (0.643) | 0.998 (0.663) |
| Completeness (spherical) (%) | 55.7 (8.9) | 67.2 (13.6) | 57.8 (10.1) |
| Completeness (ellipsoidal) (%) | 96.1 (76.7) | 95.2 (77.8) | 95.3 (79.5) |
| Redundancy | 21.1 (20.4) | 20.0 (20.4) | 21.2 (22.1) |
| **Refinement** |  |  |  |
| Resolution (Å) | 32.21 – 1.67 | 43.78–2.29 | 32.17–1.98 |
| No. reflections | 70727 | 32659 | 43339 |
| $R_{work} / R_{free}$ | 0.1658/0.1896 | 0.1847/0.2183 | 0.1733/0.2108 |
| Cruickshank DPI (Å) | 0.2 | 0.2 | 0.2 |
| No. atoms | 5547 | 5192 | 5275 |
| Protein | 4961 | 4960 | 4939 |
| Ligand/ion | 4 | 4 | 4 |
| Water | 582 | 228 | 332 |
| *B*-factors | 40.52 | 46.62 | 45.50 |
| Protein | 40.10 | 46.84 | 45.63 |
| Ligand/ion | 27.57 | 35.65 | 52.63 |
| Water | 44.26 | 42.10 | 43.40 |
| R.m.s. deviations |  |  |  |
| Bond lengths (Å) | 0.014 | 0.004 | 0.008 |
| Bond angles (°) | 1.296 | 0.731 | 1.096 |

ᵃValues in parentheses are for highest-resolution shell.

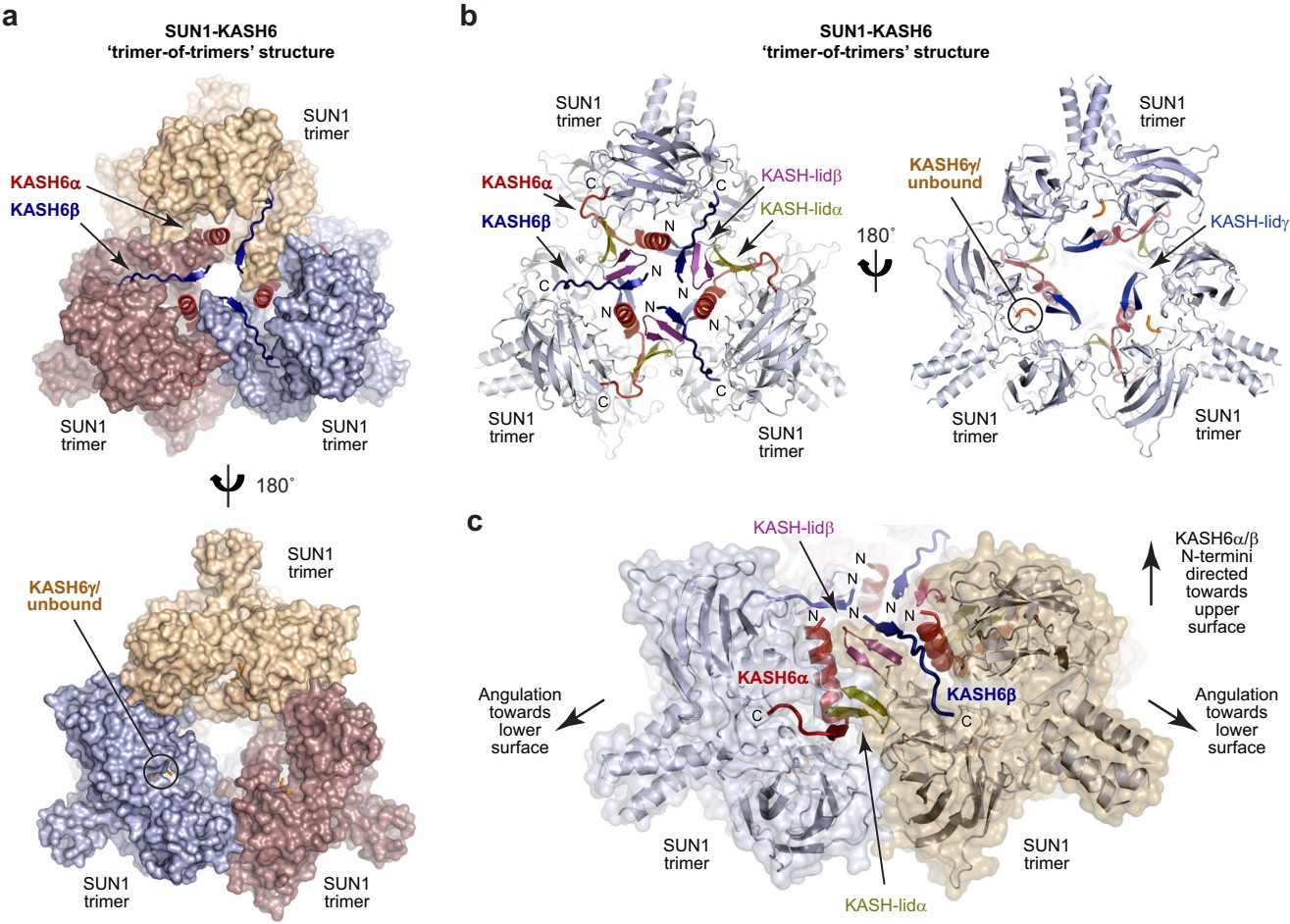

**Fig. 2 Crystal structure of SUN1-KASH6. a–c** Crystal structure of SUN1-KASH6 (1.7 Å resolution) in a 'trimer-of-trimers' configuration. **a, b** Overall structure with SUN domains shown as (**a**) surfaces and (**b**) cartoons. The KASH6α and KASH6β molecules bound to the three SUN1 trimers interact extensive with each other, and their associated KASH-lids at the centre of the structure, and emerge on the upper molecular surface. **c** The trimer-of-trimers interface is mediated by interactions between KASH6α and KASH6β molecules, and their associated KASH-lids, from adjacent SUN1 trimers. In this interface, KASH-lidα and KASH-lidβ are sandwiched between KASH6α and KASH6β in a six-stranded β-sheet. The N-termini of KASH6α and KASH6β are directed towards the upper surface of the molecule, whilst SUN1 trimers are angled with their N-terminal trimer coiled-coiled pointing downwards towards the lower surface of the molecule.

neighbouring trimers (Fig. 2a-c). KASH6 peptides are clearly bound at both of these KASH-binding pockets of each SUN1 trimer (these are herein referred to as KASH6α and KASH6β). The third KASH-binding pocket of each SUN1 trimer is located on the underside of the trimer-of-trimers structure, away from inter-trimer interfaces (Fig. 2a, b). Here, we observed poor peptide electron density in the 1.7 Å resolution structure, and could build only three C-terminal KASH6 amino-acids (herein referred to as KASH6γ), with 71% occupancy (Supplementary Fig. 2a). In other crystal structures, at resolutions of 2.0 Å and 2.3 Å, we observed no third peptide, with the KASH-binding pocket containing only water molecules (Supplementary Fig. 2b). In absence of KASH6γ, the remaining structure was essentially unaltered (r.m.s. deviation = 0.1 Å). Hence, the third binding pocket on the lower surface of the trimer-of-trimers structure cannot stably bind to a KASH6 peptide. We conclude that the SUN1-KASH6 structure is a 9:6 complex in which three SUN1 trimers are bound to six KASH6 peptides.

The KASH6α and KASH6β peptides bound to each SUN1 trimer adopt different conformations. These are arranged such that each inter-trimer interface is formed of intricate interactions between KASH6α and KASH6β peptides, and the associated KASH-lids, of neighbouring SUN1 trimers (Fig. 2c). Further,

their distinct conformations mean that all KASH6α and KASH6β peptides of the structure are oriented with their N-termini directed towards the top surface of the molecule (Fig. 2a-c). This asymmetry within the SUN1-KASH6 trimer-of-trimers structure positions all six stably bound KASH6 peptides in a manner that is suitable for the insertion of their immediately upstream transmembrane helices into the outer nuclear membrane. Hence, uniquely amongst SUN-KASH crystal structures, the architecture of the asymmetric SUN1-KASH6 trimer-of-trimers structure appears to be compatible with the steric requirements of membrane insertion, and thereby of LINC complex assembly in vivo.

**Distinct KASH domain conformations within the SUN1-KASH6 structure.** In previous crystal structures of SUN-KASH complexes, the beta-hairpin KASH-lids that extend from SUN domains have retained the same orientation despite differences in the structures adopted by divergent KASH domains[13,34,37,38]. In contrast, the distinct KASH6 conformations observed in the SUN1-KASH6 structure are associated with differences in the orientation of their associated KASH-lids (Supplementary Fig. 3). The KASH-lid associated with KASH6α adopts the canonical

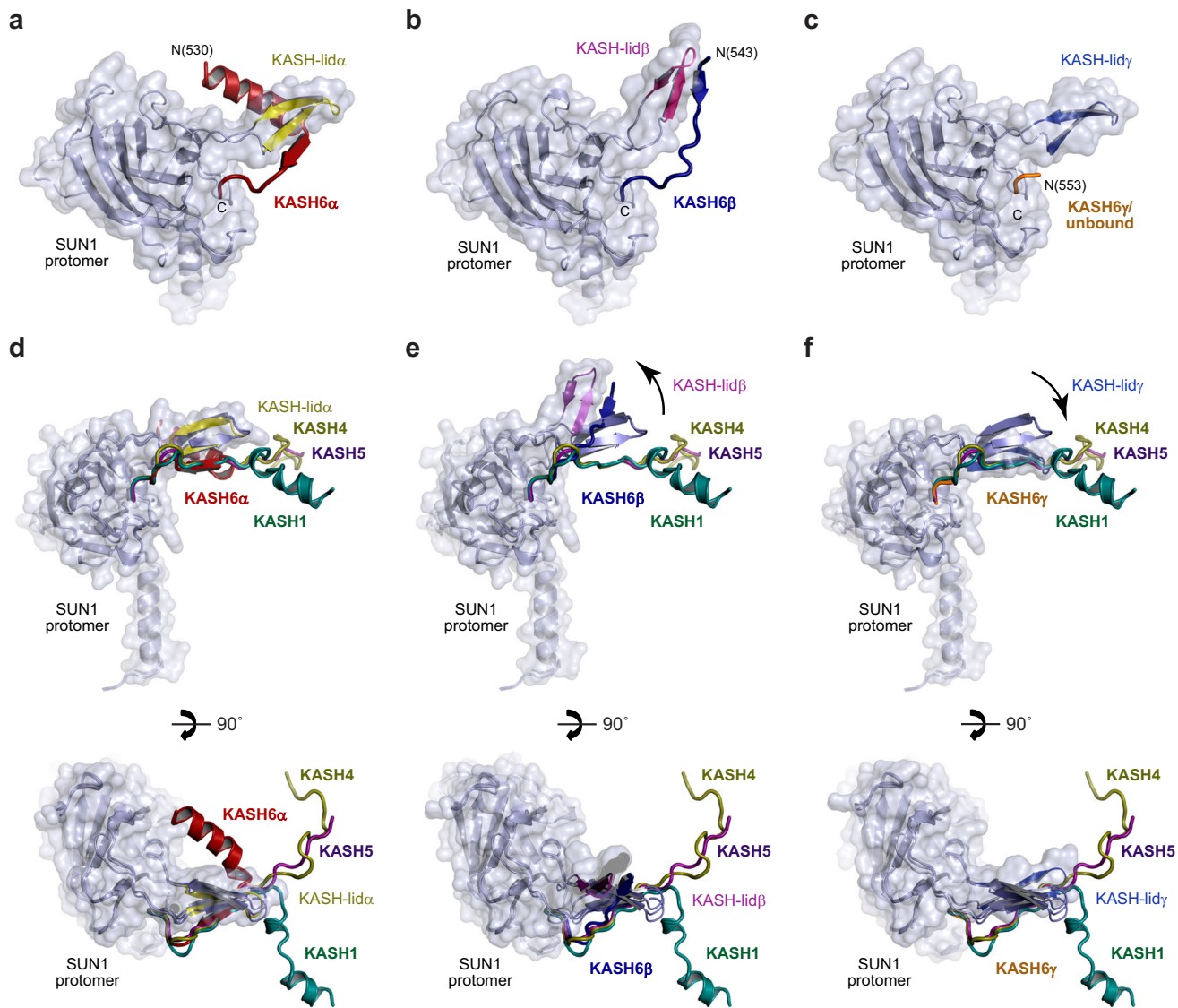

**Fig. 3 Unique conformations of KASH6 peptides within the SUN1-KASH6 structure. a–c** Alternative conformations adopted by KASH6 peptides and their associated SUN domain KASH-lids for (**a**) KASH6α, (**b**) KASH6β and (**c**) KASH6γ. **d–f** Superposition of the KASH peptide and its associated SUN1 protomer from previous crystal structures of SUN1-KASH1 (green; PDB accession 6R15)[37], SUN1-KASH4 (brown; PDB accession 6R16)[37] and SUN1-KASH5 (purple; PDB accession 6R2I)[37], with (**d**) SUN1-KASH6α (red), (**e**) SUN1-KASH6β (blue) and (**f**) SUN1-KASH6γ (yellow). **d** KASH6α follows the same initial path, undergoing β-interaction with its canonically-oriented KASH-lid, as in all previous SUN1-KASH structures. It then hooks below KASH-lidα, undergoing a 90° bend to form an α-helix that is oriented in the opposite direction to the partial α-helical end of KASH1. **e** KASH6β forms the same β-interaction with its KASH-lid as in all other SUN1-KASH structures, but with an unusually high angulation of both KASH6β and KASH6-lidβ. **f** KASH6γ was only observed as three amino-acids at the KASH-binding pocket, with partial occupancy, and in only a subset of crystal structures. In absence of the β-interaction, its KASH-lid adopts a low angulation in which its first β-strand overlays with the KASH-lid-interacting β-strand of other KASH peptides.

conformation (Fig. 3a). In contrast, the KASH-lid associated with KASH6β has a high angulation (Fig. 3b), and the KASH-lid at the KASH6γ/unbound site has a low angulation (Fig. 3c). These are herein referred to as KASH-lidα, KASH-lidβ and KASH-lidγ, respectively.

The KASH6α peptide follows the canonical binding mode of C-terminal insertion into the KASH-binding pocket, followed by a β-interaction with the canonical KASH-lidα (Fig. 3a)[13,34,37,38]. However, KASH6α then hooks underneath the KASH-lid, and undergoes a 90° turn to form an N-terminal α-helix that is perpendicular to the coiled-coil axis of the SUN1 trimer (Fig. 3a). This conformation resembles that of KASH1/2 structures, although the turn is in the opposite direction, such that N-terminal α-helices of KASH6α point inwards towards the centre of the SUN1 trimer, whereas those of KASH1/2

point outwards (Fig. 3d). Importantly, the three KASH6α α-helices of the trimer-of-trimers structure are parallel, with their N-termini directed towards the top surface of the molecule (Fig. 2a-c).

The KASH6β peptide also follows the canonical pattern of C-terminal insertion into the KASH-binding pocket and β-interaction with KASH-lidβ (Fig. 3b). However, it does not hook under the KASH-lid or form an N-terminal α-helix. Instead, its β-structure is extended, in a manner similar to KASH5 (Fig. 3b, e)[37,38]. The resultant structure with KASH-lidβ is unusually highly angled, directed away from the core of the SUN1 trimer (Fig. 3b). The N-termini of the three KASH6β peptides of the structure point towards the same molecular surface, and are each sandwiched between the N-terminal α-helices of two KASH6α peptides (Fig. 2a-c). Hence, the N-termini of all six KASH6

peptides are located in closely proximity, on the upper surface of the molecule.

At the remaining KASH6γ/unbound site, KASH-lidγ adopts a low angled conformation in which its β-structure follows the path that would normally be occupied by the KASH peptide (Fig. 3c, f). This localisation likely acts as a steric hindrance for KASH-binding. As the only remaining binding interface is the minimal binding pocket for the KASH peptide C-terminus, this likely substantially reduces the binding affinity, accounting for the lack of stable binding by a third KASH6 peptide.

**KASH domains and KASH-lids form extensive inter-trimer interfaces**. The inter-trimer interfaces that hold together the SUN1-KASH6 structure are formed by KASH6α and KASH6β molecules of neighbouring SUN1 trimers interacting via their associated KASH-lids (Fig. 2c). At its core, the inter-trimer interface is formed of a six-stranded β-sheet, in which KASH-lidα and KASH-lidβ of adjacent SUN1 trimers interact together, sandwiched between the β-strands of their KASH6α and KASH6β peptides (Fig. 2c). This interaction mode resembles the KASH-mediated head-to-head interface of the SUN1-KASH5 6:6 structure[37,38]. The inter-trimers interface is stabilised by hydrophobic interactions between the surface of the β-sheet (including SUN1 amino-acids F671, I673, L675 and W676) and the N-terminal α-helix of KASH6α (including KASH6 amino-acids W534, L537 and L541) (Fig. 4a). It is supported by inter-actions of the unbound KASH-lidγ, including binding of its tip amino-acids F671 and I673 with an opposing KASH-lidα (Fig. 4b), similar to the head-to-head interface of the SUN1-KASH1 6:6 structure[37,38]. Further, KASH-lidγ interactions position SUN1 protomers with their coiled-coils angled downwards, away from the upper surface where the N-termini of KASH6 molecules present (Fig. 2c). Hence, intricate interactions between KASH6 molecules and KASH-lids of adjacent SUN1 trimers form extensive inter-trimer interfaces that establish a trimer-of-trimers platform in which KASH6 molecules are directed to the upper surface and SUN1 coiled-coils are angled towards the lower surface. This architecture appears ideal for its known biological location in the nuclear lumen, with immediately upstream KASH6 sequences inserted into the outer nuclear membrane.

**SUN1-KASH6 undergoes trimer-of-trimers assembly in solution**. Does SUN1-KASH6 form the same trimer-of-trimers structure in solution? We utilised size-exclusion chromatography coupled to multi-angle light scattering (SEC-MALS) to determine its solution oligomeric state. SEC-MALS analysis revealed that SUN1-KASH6 exists predominantly as a 6:6 complex (150 kDa) (Fig. 5a), similar to other SUN-KASH complexes[37]. As all diffracting crystals were obtained in acidic conditions, we wondered whether trimer-of-trimers formation may occur at low pH. Hence, we treated SUN1-KASH6 in mildly acidic (pH 5.0) conditions. Upon subsequent SEC-MALS analysis at pH 7.5, we observed the presence of roughly equal amounts of a 9:6 complex (212 kDa) and a 6:6 complex (150 kDa) (Fig. 5a). This transition became almost complete when we truncated the N-terminal end of the KASH6 peptide to match only the amino-acids observed in the crystal structure (Fig. 5b). This is likely due to the steric effects of removing unstructured sequence. We also observed pH-triggered 9:6 formation using the lowest detectable protein loading concentration of 0.1 mg/ml (corresponding to less than 100 nM following dilution over SEC), so trimer-of-trimers assembly occurs with high affinity (Supplementary Fig. 4). As the 9:6 complex remained intact when analysed at pH 7.5, our data suggest that treatment in mildly acidic conditions triggers an irreversible conformational change from 6:6 to 9:6 conformation.

We next assessed the solution structure of the SUN1-KASH6 9:6 and 6:6 complexes by size-exclusion chromatography coupled to small angle X-ray scattering (SEC-SAXS). The SAXS scattering curve of the 9:6 complex was closely fitted by the SUN1-KASH6 trimer-of-trimers 9:6 crystal structure ($\chi^2 = 2.22$) (Fig. 5c and Supplementary Fig. 5a, b), but not the previous SUN1-KASH5 head-to-head 6:6 crystal structure ($\chi^2 = 62.5$). In contrast, the SAXS scattering curve of the 6:6 complex was closely fitted by the SUN1-KASH5 head-to-head 6:6 crystal structure ($\chi^2 = 1.32$) (Fig. 5c and Supplementary Fig. 5a, b), but not the SUN1-KASH6 trimer-of-trimers 9:6 crystal structure ($\chi^2 = 24.5$). Further, SAXS ab initio envelopes correctly predicted the trimer-of-trimers and head-to-head conformations of 9:6 and 6:6 species, aligning closely to docked SUN1-KASH6 and SUN1-KASH5 crystal structures (Fig. 5d). To confirm these findings, we directly visualised the SUN1-KASH6 core complex following treatment at pH 5.0 by cryo-electron microscopy (cryo-EM). Here, reference-free 2D class averages revealed trimer-of-trimers and head-to-head assemblies that closely correspond to the 9:6 structure reported herein and previously observed 6:6 structures, respectively (Fig. 5e).

Together, these orthogonal approaches strongly support that the SUN1-KASH6 9:6 complex in solution has the same trimer-of-trimers configuration that is observed in the crystal structure, and that the 6:6 complex in solution has the same head-to-head structure that was previously observed for other SUN-KASH complexes. Hence, treatment at pH 5.0 triggers the conformational change of SUN1-KASH6 from a symmetric head-to-head 6:6 structure into an asymmetric trimer-of-trimers 9:6 assembly (Fig. 5f). This conformational change requires the dissociation of two KASH6 molecules from each 6:6 complex. Hence, the few amino-acids of a KASH6γ peptide observed in the 1.7 Å crystal structure were likely remnants of the peptide having failed to fully dissociate upon trimer-of-trimers assembly.

**The KASH6 N-terminal α-helix mediates trimer-of-trimers assembly**. What are the molecular determinants of SUN1-KASH6 trimer-of-trimers assembly? The most unique structural feature of SUN1-KASH6 is the N-terminal α-helix of KASH6. In the KASH6α conformation, the N-terminal α-helix hooks under the KASH-lid and is oriented towards the centre of the SUN1 trimer (Figs. 2a, b and 3a, d). This establishes an asymmetry by sterically blocking other KASH6 peptides of the same trimer from adopting the same configuration. In this central position, it mediates both inter- and intra-trimer interactions (Fig. 4a, b). Hence, the N-terminal α-helix of KASH6 appears to co-ordinate interactions that hold together the asymmetric trimer-of-trimers structure. To test this, we introduced point mutations of key amino-acids within this and a downstream region (Fig. 6a-c).

At the inter-trimer interface, KASH6 amino-acid W534 forms hydrophobic contacts with KASH-lidβ of the opposing trimer, and co-ordinates a chloride ion with KASH-lidα of the same trimer (Fig. 6a). The point mutation W534A blocked pH-triggered trimer-of-trimers assembly and locked the complex in a 6:6 conformation, confirming that these interactions are essential for assembly (Fig. 6d). At the intra-trimer interface, KASH6 amino-acid H539 interacts with KASH-lidγ of the same trimer, including via a hydrogen-bond with a carbonyl group of the beta-sheet (Fig. 6b). We reasoned that its protonation may strengthen the interaction by establishing a salt bridge with the KASH-lid backbone, which we aimed to mimic constitutively by mutation to lysine. Accordingly, the H539K mutant formed 9:6 complexes in absence of treatment at pH 5.0 (Fig. 6e). In contrary, an H539Q mutant, which should form the same hydrogen bond but without the ability for protonation, failed to undergo trimer-of-trimers

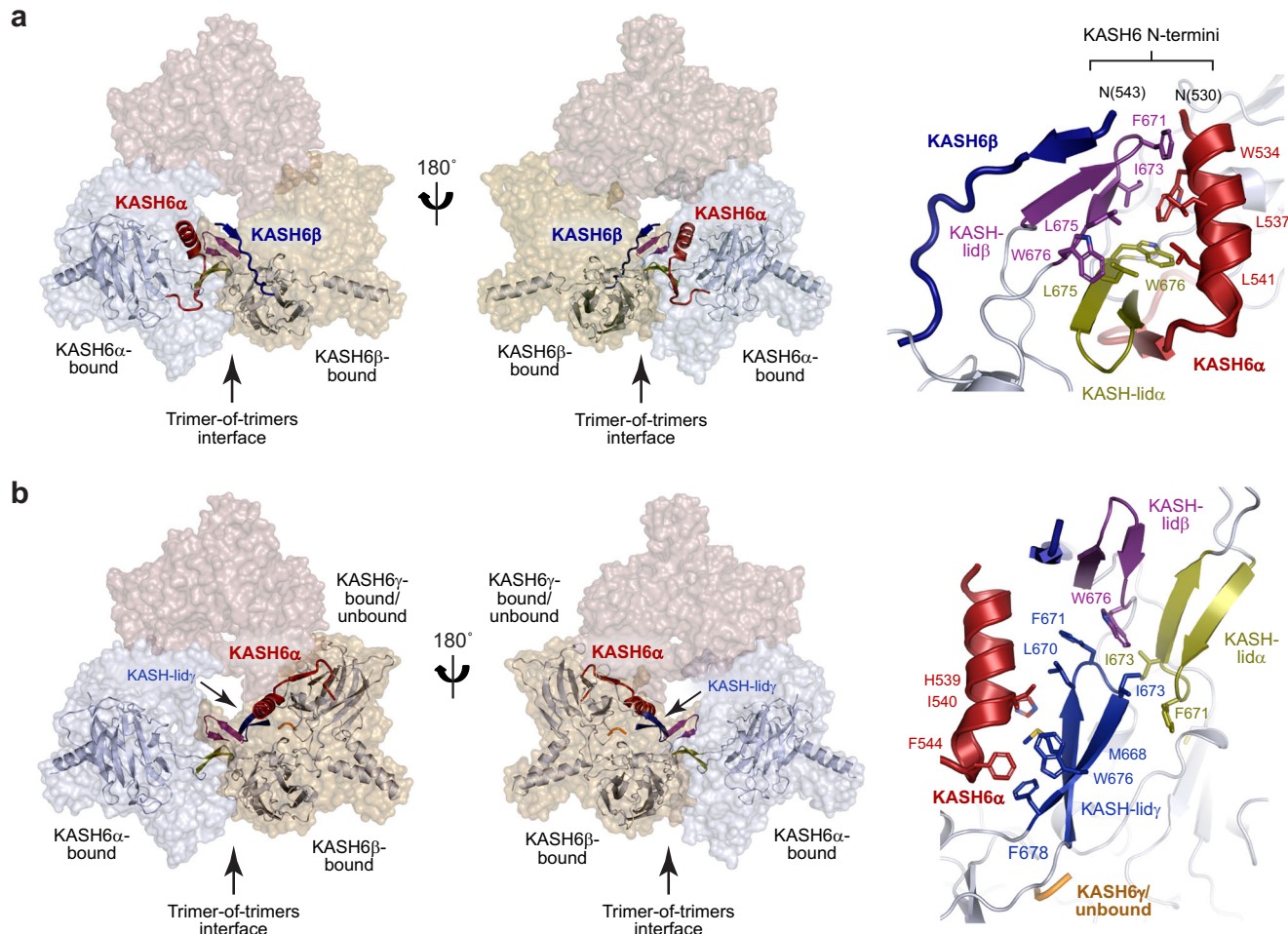

**Fig. 4 Structural roles of KASH6 peptides and KASH-lids within the trimer-of-trimers assembly. a, b** Location of interacting KASH6 peptides and associated SUN domains within the trimer-of-trimers structure (left), and their molecular details (right). (**a**) The trimer-of-trimers interface is mediated by an interaction between KASH6α (and associated SUN domain KASH-lidα) and KASH6β (and associated SUN domain KASH-lidβ) of adjacent SUN1 trimers. The KASH6α and KASH6β peptides sandwich their KASH-lids together in a β-sheet structure that is stabilised by hydrophobic packing between the KASH6α helix and the β-sheet. This results in the N-termini of KASH6α and KASH6β peptides being oriented close together on the upper molecular surface. **b** KASH-lidγ of the KASH6γ-associated or unbound SUN1 protomer interacts with the KASH6α helix and KASH-lidβ of the same SUN1 trimer, and KASH-lidα of the adjacent SUN1 trimer. Its interaction with KASH-lidβ involves the same molecular contacts as the head-to-head interfaces of SUN1-KASH1 and SUN1-KASH5 6:6 complexes[37].

assembly following acidic treatment, remaining as a stable 6:6 complex (Fig. 6f). Hence, the N-terminal α-helix of KASH6 is critical for trimer-of-trimers assembly. In pH-triggered assembly, we propose that H539 protonation binds the α-helix to the SUN1 trimer in a conformation conducive with the establishment of inter-trimer interactions by W534.

KASH6 contains a second histidine residue, H549, which is located at the hinge between the KASH peptide C-terminal insertion and the β-interaction with the KASH-lid (Fig. 6c). This amino-acid is conserved as a tyrosine residue in other KASH sequences, so we wondered whether its protonation may also contribute to trimer-of-trimers assembly. Accordingly, point mutation H549Y partially blocked pH-triggered assembly, with the formation of 9:6 complexes at a proportion of approximately 45% (Fig. 6g), in contrast to 85% for the wild-type protein (Fig. 5b). Thus, protonation of H549 contributes to, but is not essential for, pH-triggered trimer-of-trimers assembly of SUN1-KASH6.

In conclusion, SUN1-KASH6 forms an asymmetric trimer-of-trimers structure, both in crystals and in solution, in which bound KASH peptides emanate on the top surface of the molecule. This

architecture is theoretically compatible with the steric requirements for insertion of upstream transmembrane helices of all bound KASH molecules into the outer nuclear membrane (Fig. 7).

## Discussion

The SUN1-KASH6 9:6 crystal structure provides a distinct snapshot of the molecular architecture that may be adopted by a LINC complex. In this structure, SUN1 trimers are arranged around a three-fold symmetry axis, such that their upstream trimeric coiled-coils emanate at the points of a triangle. In keeping with previous SUN-KASH structures, KASH6 C-termini are bound in pockets between adjacent SUN1 protomers[13,34,37,38]. However, whilst the two binding pockets at the inter-trimer interface were occupied with KASH6 peptides, the third pocket on the lower surface of the molecule was either unoccupied or contained only a few C-terminal amino-acids. Hence, the third pocket does not stably bind KASH6 peptides. The two robustly bound KASH6 peptides of each SUN1 trimer adopt distinct conformations from each other, and from other

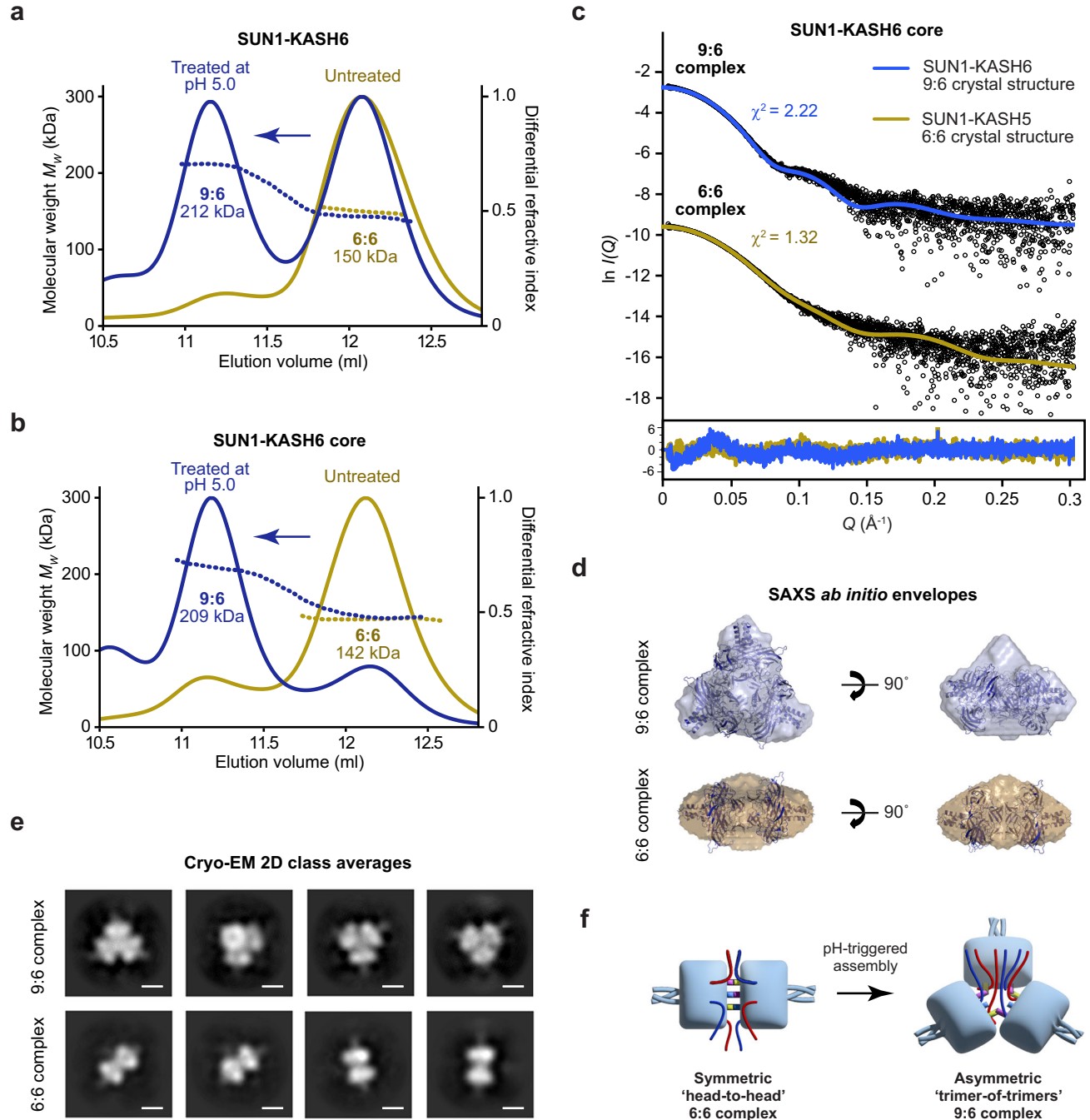

**Fig. 5 Assembly of SUN1-KASH6 9:6 complexes in solution. a, b** Size-exclusion chromatography coupled to multi-angle light scattering showing that (**a**) SUN1-KASH6 and (**b**) SUN1-KASH6 core are 6:6 complexes that assemble into 9:6 complexes following treatment at pH 5.0 (theoretical 9:6–232 kDa and 222 kDa; theoretical 6:6–165 kDa and 155 kDa). **c, d** Small-angle X-ray scattering of SUN1-KASH6 core 9:6 and 6:6 species. **c** SAXS scattering data overlaid with the theoretical scattering curves of the 9:6 and 6:6 crystal structures of SUN1-KASH6 and SUN1-KASH5 (PDB accession 6R2I)[37], showing $\chi^2$ values of 2.22 and 1.32, respectively (for other combinations, $\chi^2 > 24$). Source data are available in Supplementary Data 1. (**d**) SAXS ab initio models of SUN1-KASH6 core 9:6 and 6:6 species. Filtered averaged models from 20 independent *DAMMIF* runs are shown with the crystal structures of SUN1-KASH6 and SUN1-KASH5 docked into the respective SAXS envelopes. **e** Cryo-EM reference-free 2D class averages of SUN1-KASH6, corresponding to the 9:6 complex (top) and 6:6 complex (bottom). Scale bars, 50 Å. **f** SUN1-KASH6 forms a symmetric head-to-head 6:6 complex in solution that undergoes pH-triggered conformational change into an asymmetric trimer-of-trimers 9:6 complex.

KASH domains[13,34,37,38]. KASH6α undergoes a β-interaction with its associated KASH-lid, which it hooks underneath, and then forms an α-helix perpendicular to the SUN1 trimer. In contrast, KASH6β forms a more extensive β-interaction with its KASH-lid. At each inter-trimer interface, KASH6α and KASH6β from interacting SUN1 trimers form integrated structures, in which their associated KASH-lids are sandwiched within a six-stranded β-sheet. Hence, this highly intertwined structure is stabilised by extensive interactions between adjacent SUN1 trimers and KASH peptides. Importantly, the distinct KASH6α and KASH6β conformations result in an asymmetrical structure in which the N-termini of all six KASH6 peptides emanate towards

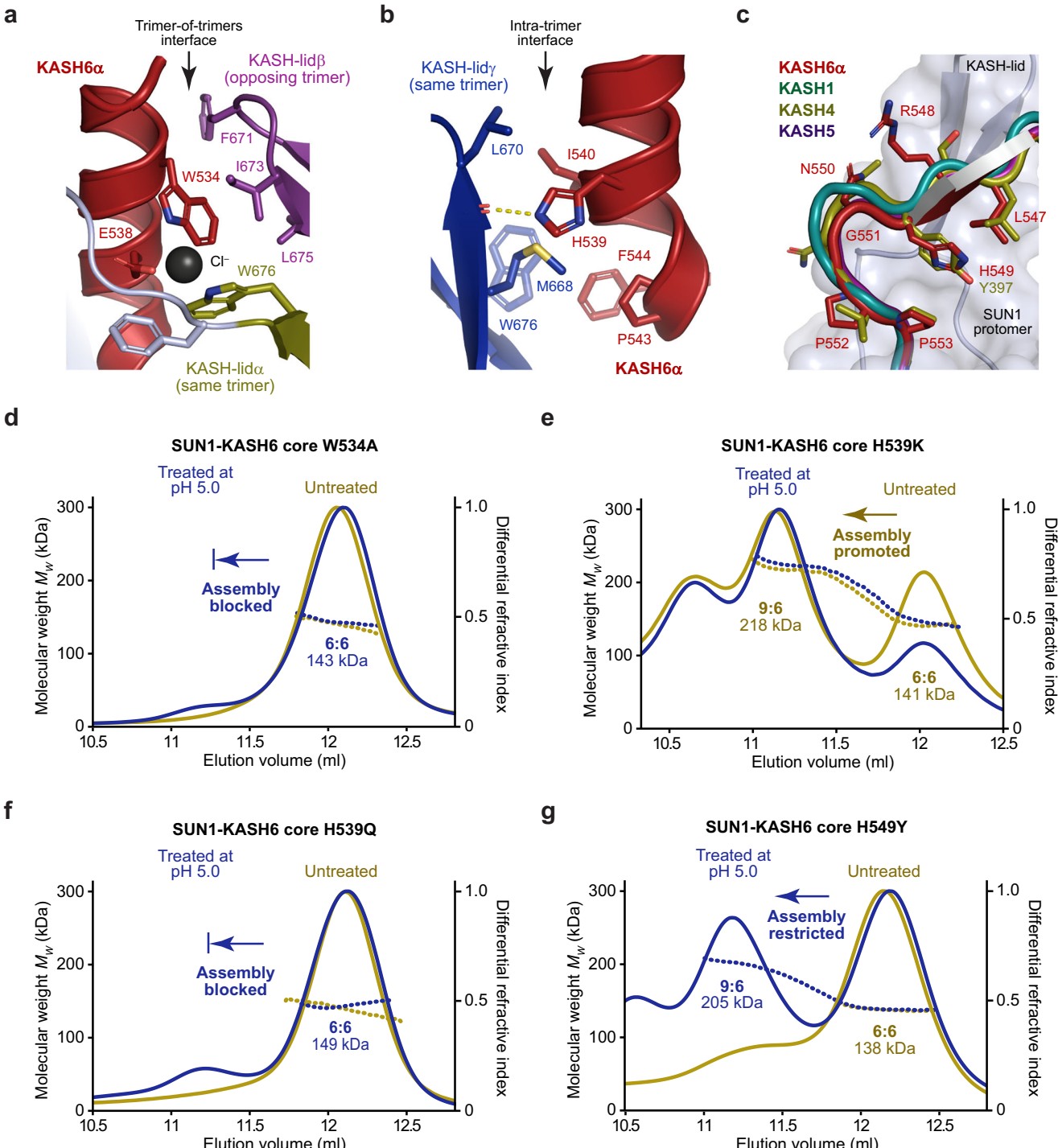

the top surface of the molecule. This configuration may favour insertion of upstream sequences of all bound KASH6 molecules into the outer nuclear membrane. Hence, uniquely amongst SUN-KASH crystal structures, the SUN1-KASH6 structure is theoretically compatible with outer nuclear membrane insertion (Fig. 7).

In solution, SUN1-KASH6 adopted the same 6:6 complex as other SUN-KASH complexes[37], but formed the 9:6 complex observed in the crystal structure upon treatment in mildly acidic conditions. This occurs through protonation of H539, which likely establishes a salt bridge with an adjacent KASH-lid that binds the N-terminal α-helix of KASH6α across the SUN1 trimer. This sterically blocks other KASH6 peptides of the same trimer

from adopting the same configuration, and establishes inter-trimer interactions by W534. Hence, the N-terminal α-helix co-ordinates interactions that hold together the asymmetric trimer-of-trimers structure. Additionally, protonation of H549, which is located at the hinge between the KASH peptide C-terminal insertion and the β-interaction with the KASH-lid, also contributes to assembly. An intriguing feature of pH-triggered 9:6 assembly is that it requires dissociation of two KASH6 molecules from each 6:6 complex. This occurs due to steric hindrance from the low-angled conformation of KASH-lidγ, which leaves only a minimal pocket available for binding to the KASH peptide C-terminus. An advantage of losing the third KASH6 peptide is that there is no simple topology in which it could reach the same

**Fig. 6 Molecular determinants of SUN1-KASH6 trimer-of-trimers assembly. a–c** Structural details of interactions formed by KASH6α amino-acids W534, H539 and H549. **a** W534 mediates inter-trimer interactions of KASH6α (red) by forming hydrophobic interactions with F671, I673 and L675 of KASH-lidβ from the opposing trimer (purple). It also co-ordinates a chloride ion along with E538 of KASH6α and W676 of KASH-lidα from the same trimer (yellow). **b** H539 interacts with a backbone carbonyl of KASH-lidγ from the same trimer (blue), which may constitute a salt bridge upon histidine protonation. This interface also includes interactions between H539, I540, P534 and F544 of KASH6α (red) and W676, M668 and L670 of KASH-lidγ from the same trimer (blue). **c** Superposition of KASH6α (red) and its associated SUN1 protomer with previous crystal structures (KASH1, green; KASH4, brown; KASH5, purple). KASH6α amino-acid H549 differs from other KASH proteins that have a tyrosine at this position (for clarity, only Y397 of KASH4 is shown). **d–g** Size-exclusion chromatography coupled to multi-angle light scattering of point mutants of the SUN1-KASH6 core complex in which samples were analysed in native conditions (yellow) and following treatment at pH 5.0 (blue). (**d**) W534A mutation largely blocked assembly, remaining in a 6:6 complex (143 kDa) following acidic treatment. **e** H539K mutation promoted assembly in the absence of acidic treatment, with untreated material forming 9:6 and 6:6 complexes (218 kDa and 141 kDa) at approximately 60% and 40% (by mass), respectively. **f** H539Q mutation largely blocked assembly, remaining in a 6:6 complex (149 kDa) following acidic treatment. **g** H549Y mutation partially blocked assembly, with treated material forming 9:6 and 6:6 complexes (205 kDa and 138 kDa) at approximately 45% and 55% (by mass), respectively. For comparison, the treated wild-type complex (Fig. 5a) formed 9:6 and 6:6 complexes at approximately 85% and 15% (by mass), respectively.

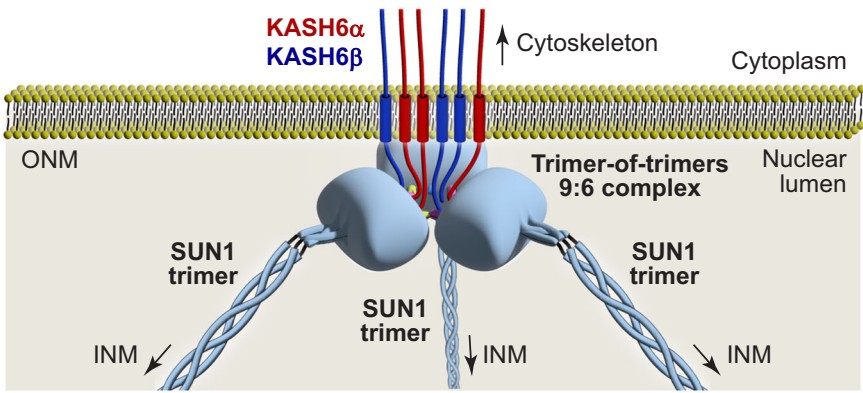

**Fig. 7 Model of a SUN1-KASH6 9:6 LINC complex.** Schematic model of how a SUN1-KASH6 asymmetric trimer-of-trimers LINC complex may be positioned with the nuclear lumen. The six bound KASH6 domains emerge from the 9:6 complex on the top surface of the molecule, favouring insertion of their immediately upstream transmembrane helices into the outer nuclear membrane (ONM). The SUN1 trimers are tilted downwards within the trimer-of-trimers structure, facilitating the angled passage of SUN1's upstream trimeric coiled-coil across the nuclear lumen, towards the inner nuclear membrane (INM).

side of the molecule as the other peptides. Hence, this appears to be favourable for the formation of a structure in which all bound KASH6 peptides can be inserted into a biological membrane located on one side of the molecule.

How does the pH-triggered assembly observed in vitro relate to the likely mechanism of SUN-KASH6 assembly in vivo? As the 9:6 complex remained stable at pH 7.5, low pH appeared to overcome an energy barrier, enabling transition from head-to-head to trimer-of-trimers structure. We envisage several possibilities for how this may occur in vivo. Firstly, the presence of chaperones in mammalian cells may favour the formation of the 9:6 complex. Secondly, an interacting protein may protonate the H539 residue. Thirdly, a local pH in the range of 5.0 may protonate H539 and trigger assembly of the 9:6 complex. Whilst the pH within the nuclear lumen is likely to be close to neutral, the negative charge of lipid membranes attracts a proton cloud, reducing the local pH at the membrane interface to several units lower than the bulk pH of the compartment[39]. Hence, the SUN-KASH complex may undergo trimer-of-trimers assembly upon membrane insertion owing to a low local pH at the interface of the outer nuclear membrane.

What is the structure of the LINC complex in vivo? The only reliable answer to this question will come from its direct visualisation in situ by cryo-electron microscopy. However, current technologies are not yet sufficient to resolve narrow coiled-coils and small complexes associated with membrane surfaces within cells. Hence, we lack critical information regarding the stoichiometry of individual SUN-KASH complexes in vivo and the

structural consequences of membrane insertion. Thus, we are currently limited to inferring its possible cellular architecture based on crystal structures, in vitro biophysical data and the inherent geometries associated with an assumed planar membrane. On this basis, the assembly of SUN1 and KASH6 molecules within the 9:6 trimer-of-trimers structure is theoretically compatible with outer nuclear membrane insertion (Fig. 7). We did not observe 9:6 assembly upon acidic treatment of SUN1-KASH1, SUN1-KASH4 and SUN1-KASH5, so our findings may be specific to the non-canonical KASH6 sequence. Nevertheless, the wider principle of higher-order asymmetric assembly, owing to a trigger or the steric effects of membrane proximity/insertion, may be broadly applicable to LINC complexes. Indeed, we previously reported that SUN1-KASH4 formed a 12:12 species in addition to its predominant 6:6 complex[37]. Further, different LINC complex architectures may form in different cells, at different times, and in response to different stresses[3]. Hence, to explain the varied and dynamic roles of LINC complexes in cells, we must consider all possible architectures that have been hypothesised and observed, including 3:3 linear complexes[13,34], hinged 6:6 complexes[37], and the newly observed asymmetric 9:6 architecture of SUN1-KASH6.

Overall, the SUN1-KASH6 structure reported herein establishes three important precedents. Firstly, stable SUN-KASH LINC complexes can include unoccupied KASH-binding pockets. Secondly, the same KASH peptide, and associated KASH-lid, can adopt alternative conformations within the same SUN-KASH complex. Finally, SUN-KASH complexes can form asymmetric

assemblies in which all bound KASH peptides emerge on the same surface of the complex. Hence, it is possible for a LINC complex to assemble in which all bound KASH molecules are oriented in a manner suitable for insertion into the outer nuclear membrane (Fig. 7). Whilst our findings relate to SUN1-KASH6, it is possible that other LINC complexes may undergo conformational change into similar higher order assemblies for membrane insertion. Indeed, the trimer-of-trimers interface of SUN1-KASH6 involves the hinging motion that we previously observed for other SUN-KASH molecules[37]. Thus, we provide the first reported case in which an experimental structure of a LINC core complex, and its solution biophysical data, are theoretically compatible with outer nuclear membrane insertion.

## Methods

**Recombinant protein expression and purification.** The SUN domain sequence of human SUN1 (amino-acids 616-812) was fused to a TEV-cleavable N-terminal His₆-GCN4 tag and cloned into a pRSF-Duet1 vector (Merck Millipore). KASH domain sequences of human JAW1/LRMP (amino-acids 515-555 and 531-555 for KASH6 and KASH6 core, respectively; for wild-type and point mutations W534A, H539Q, H539K and H549Y) were cloned into pMAT11 vectors[40] for expression with N-terminal His₆-MBP tags. SUN1 and KASH6 were co-expressed in BL21 (DE3) cells (Novagen) in 2xYT media, and induced at an OD of 0.8 with 0.5 mM IPTG for 16 hours at 25 °C. Cells were lysed by cell disruption in 20 mM Hepes (pH 7.5), 500 mM KCl, and lysate was purified through consecutive Ni-NTA (Qiagen), amylose (NEB) and HiTrap Q HP (GE Healthcare) anion exchange chromatography. Fusion proteins were cleaved by TEV protease, and cleaved samples were purified by HiTrap Q HP anion exchange, followed by size-exclusion chromatography (HiLoad 10/300 Superdex 200, GE Healthcare) in 20 mM Hepes (pH 7.5), 150 mM KCl, 2 mM DTT. Protein samples were concentrated using Amicon Ultra-4 10 MWCO Centrifugal devices (Thermo Scientific), and flash frozen in liquid nitrogen and stored in -80 °C. Protein samples were analysed by SDS-PAGE with Coomassie staining, and concentrations were determined by UV spectroscopy using a Cary 60 UV spectrophotometer (Agilent) with extinction coefficients and molecular weights calculated by ProtParam (http://web.expasy.org/protparam/).

**Crystallisation and structure solution of SUN1-KASH6 (1.7 Å) at 9:9 stoichiometry.** SUN1-KASH6 protein crystals were obtained through vapour diffusion in sitting drops, by mixing protein at 12 mg/ml with crystallisation solution (0.15 M Magnesium chloride, 0.1 M MES pH 6.0, 3% w/v PEG 6000) and equilibrating at 20 °C. Crystals were cryo-protected by addition of 25% ethylene glycol, and were cryo-cooled in liquid nitrogen. X-ray diffraction data were collected at 0.6702 Å, 100 K, as 3600 consecutive 0.10° frames of 0.005 s exposure on a Pilatus3 6 M detector at beamline I24 of the Diamond Light Source synchrotron facility (Oxfordshire, UK). Data were processed using *AutoPROC*[41], in which indexing, integration, scaling were performed by *XDS*[42] and *Aimless*[43], and anisotropic correction with a local $I/\sigma(I)$ cut-off of 1.2 was performed by *STARANISO*[44]. Crystals belong to hexagonal spacegroup P6₃ (cell dimensions a = 134.09 Å, b = 134.09 Å, c = 106.59 Å, α = 90°, β = 90°, γ = 120°), with one 3:3 SUN1-KASH6 complex in the asymmetric unit. Structure solution was achieved by molecular replacement using *PHASER*[45], in which three copies of the SUN domain from the SUN1-KASH5 structure (PDB accession 6R2I)[37] were placed in the asymmetric unit. Model building was performed through iterative re-building by *PHENIX Autobuild*[46] and manual building in *Coot*[47]. The structure was refined using *PHENIX refine*[46],

using isotropic (water) atomic displacement parameters, with 22 TLS groups. The structure was refined against data to anisotropy-corrected data with resolution limits between 1.67 Å and 2.35 Å, to *R* and $R_{free}$ values of 0.1658 and 0.1896, respectively, with 96.41% of residues within the favoured regions of the Ramachandran plot (0 outliers), clashscore of 7.38 and overall *MolProbity* score of 1.64[48]. The final SUN1-KASH6 model was analysed using theOnline DPI webserver (http://cluster.physics.iisc.ernet.in/dpi) to determine a Cruikshank diffraction precision index (DPI) of 0.2 Å[49].

**Crystallisation and structure solution of SUN1-KASH6 (2.3 Å) at 9:6 stoichiometry.** SUN1-KASH6 protein crystals were obtained through vapour diffusion in sitting drops, by mixing protein at 12 mg/ml with crystallisation solution (0.1 M Bis Tris pH 5.5, 0.3 M magnesium formate) and equilibrating at 20 °C. Crystals were cryo-protected by addition of 30% ethylene glycol, and were cryo-cooled in liquid nitrogen. X-ray diffraction data were collected at 0.9999 Å, 100 K, as 3600 consecutive 0.10° frames of 0.010 s exposure on a Pilatus3 6 M detector at beamline I24 of the Diamond Light Source synchrotron facility (Oxfordshire, UK). Data were processed using *AutoPROC*[41], in which indexing, integration, scaling were performed by *XDS*[42] and *Aimless*[43], and anisotropic correction with a local $I/\sigma(I)$ cut-off of 1.2 was performed by *STARANISO*[44]. Crystals belong to hexagonal spacegroup P6₃ (cell dimensions a = 133.76 Å, b = 133.76 Å, c = 106.71 Å, α = 90°, β = 90°, γ = 120°), with one 3:2 SUN1-KASH6 complex in the asymmetric unit. Structure solution was achieved by molecular replacement using *PHASER*[45], in which three copies of the SUN domain from the SUN1-KASH5 structure (PDB accession 6R2I)[37] were placed in the asymmetric unit. Model building was performed through iterative re-building by *PHENIX Autobuild*[46] and manual building in *Coot*[47]. The structure was refined using *PHENIX refine*[46], using isotropic (water) atomic displacement parameters, with 16 TLS groups. The structure was refined against data to anisotropy-corrected data with resolution limits between 2.29 Å and 2.93 Å, to *R* and $R_{free}$ values of 0.1847 and 0.2183, respectively, with 97.06% of residues within the favoured regions of the Ramachandran plot (0 outliers), clashscore of 4.92 and overall *MolProbity* score of 1.42[48]. The final SUN1-KASH6 model was analysed using the Online_DPI webserver (http://cluster.physics.iisc.ernet.in/dpi) to determine a Cruikshank diffraction precision index (DPI) of 0.2 Å[49].

**Crystallisation and structure solution of SUN1-KASH6 (2.0 Å) at 9:6 stoichiometry.** SUN1-KASH6 protein crystals were obtained through vapour diffusion in sitting drops, by mixing protein at 12 mg/ml with crystallisation solution (0.15 M sodium acetate; 0.1 M Sodium citrate tribasic dihydrate pH 5.5) and equilibrating at 20 °C. Crystals were cryo-protected by the addition of 25% ethylene glycol, and were cryo-cooled in liquid nitrogen. X-ray diffraction data were collected at 0.6702 Å, 100 K, as 3600 consecutive 0.10° frames of 0.005 s exposure on a Pilatus3 6 M detector at beamline I24 of the Diamond Light Source synchrotron facility (Oxfordshire, UK). Data were processed using *AutoPROC*[41], in which indexing, integration, scaling were performed by *XDS*[42] and *Aimless*[43], and anisotropic correction with a local $I/\sigma(I)$ cut-off of 1.2 was performed by *STARANISO*[44]. Crystals belong to hexagonal spacegroup P6₃ (cell dimensions a = 134.09 Å, b = 134.09 Å, c = 106.59 Å, α = 90°, β = 90°, γ = 120°), with one 3:2 SUN1-KASH6 complex in the asymmetric unit. Structure solution was achieved by molecular replacement using *PHASER*[45], in which a 3:2 complex from the SUN1-KASH6 9:9 structure (PDB accession 8B46) was placed in the asymmetric unit. Model building

was performed through iterative re-building by *PHENIX Autobuild*[46] and manual building in *Coot*[47]. The structure was refined using *PHENIX refine*[46], using isotropic (water) atomic displacement parameters, with 16 TLS groups. The structure was refined against data to anisotropy-corrected data with resolution limits between 1.98 Å and 2.77 Å, to $R$ and $R_{free}$ values of 0.1733 and 0.2107, respectively, with 96.40% of residues within the favoured regions of the Ramachandran plot (0 outliers), clashscore of 6.39 and overall *MolProbity* score of 1.59[48]. The final SUN1-KASH6 model was analysed using the Online_DPI webserver (http://cluster.physics.iisc.ernet.in/dpi) to determine a Cruikshank diffraction precision index (DPI) of 0.2 Å[49].

**Size-exclusion chromatography coupled to multiangle light scattering (SEC-MALS).** The absolute molecular masses of SUN1-KASH6 complexes were determined by size-exclusion chromatography multi-angle light scattering (SEC-MALS). Protein samples at 8-13 mg/ml were loaded onto a Superdex™ 200 Increase 10/300 GL size exclusion chromatography column (GE Healthcare) in 20 mM Hepes pH 7.5, 150 mM KCl, 2 mM DTT, at 0.5 ml/min using an ÄKTA™ Pure (GE Healthcare). For induction of 9:6 complex formation, samples were treated in 20 mM Sodium acetate pH 5.0, 150 mM KCl, 10% glycerol, 2 mM DTT overnight, and were analysed by size-exclusion chromatography in the buffer described above. The column outlet was fed into a DAWN® HELEOS™ II MALS detector (Wyatt Technology), followed by an Optilab® T-rEX™ differential refractometer (Wyatt Technology). Light scattering and differential refractive index data were collected and analysed using *ASTRA®* 6 software (Wyatt Technology). Molecular weights and estimated errors were calculated across eluted peaks by extrapolation from Zimm plots using a *dn/dc* value of 0.1850 ml/g. SEC-MALS data are presented as differential refractive index (dRI) profiles, with fitted molecular weights ($M_W$) plotted across elution peaks.

**Size-exclusion chromatography coupled to small-angle X-ray scattering (SEC-SAXS).** SEC-SAXS experiments were performed at beamline B21 of the Diamond Light Source synchrotron facility (Oxfordshire, UK). Protein samples at 12-25 mg/ml were prepared as described above for SEC-MALS and were loaded onto a Superdex™ 200 Increase 10/300 GL size exclusion chromatography column (GE Healthcare) in 20 mM Hepes pH 7.5, 150 mM KCl, 2 mM DTT, at 0.5 ml/min using an Agilent 1200 HPLC system. The column outlet was fed into the experimental cell, and SAXS data were recorded at 12.4 keV, detector distance 4.014 m, in 3.0 s frames. Data were subtracted, averaged and analysed for Guinier region $Rg$ using *ScÅtter* 3.0 (http://www.bioisis.net), and $P(r)$ distributions were fitted using *PRIMUS*[50]. Ab initio modelling was performed using *DAMMIF*[51], in which 20 independent runs were performed in P3 and P32 symmetry for SUN1-KASH6 core 9:6 and 6:6 species, respectively. Models were averaged, and crystal structures of the SUN1-KASH6 9:6 complex (this study) and the SUN1-KASH5 6:6 complex (PDB accession 6R2I)[37] were docked into *DAMFILT* molecular envelopes using *SUPCOMB*[52]. Crystal structures were fitted to experimental data using *CRYSOL*[53]. The SUN1-KASH6 9:6 crystal structure fitted to 9:6 and 6:6 complex experimental data with $\chi^2$ values of 2.22 and 24.5, respectively, whilst the SUN1-KASH5 6:6 crystal structure (PDB accession 6R2I)[37] fitted to the same 9:6 and 6:6 complex experimental data with $\chi^2$ values of 62.5 and 1.32, respectively.

**Cryo-electron microscopy (cryo-EM).** R1.2/1.3 300-mesh copper grids (Quantifoil) were glow-discharged for 90 seconds prior to sample freezing. A volume of 4 µl of purified SUN1-KASH6 complex (1 mg/ml; following treatment) was dispensed onto the grid, blotted for 2 seconds and flash frozen in liquid ethane cooled with liquid nitrogen using a Vitrobot Mark IV (Thermo Fisher) operated at 4 °C and 100% humidity. Data were collected on an FEI Tecnai F20 cryo-electron microscope operated at 200 kV equipped with a K2 Summit direct electron detector (Gatan Inc.), using *SerialEM*[54]. Movies were recorded in electron-counting mode fractionating with a total exposure of 39 e/Å². A defocus range of -0.8 to -1.5 µm was used and the physical pixel size was 1.02 Å/pixel. The movies were gain normalised, motion-corrected, and dose-weighted with *MotionCo2*[55]. *CryoSPARC* v4.3.1[56] was used to import micrographs, perform CTF estimation with *CTFFIND4*[57], pick particles and generate 2D class averages.

**Protein sequence and structure analysis.** Multiple sequence alignments were generated using *Jalview*[58], and molecular structure images were generated using the *PyMOL* Molecular Graphics System, Version 2.0.4 Schrödinger, LLC.

**Statistics and reproducibility.** All biochemical and biophysical experiments were repeated at least three times with separately prepared recombinant protein material.

**Reporting summary.** Further information on research design is available in the Nature Portfolio Reporting Summary linked to this article.

### Data availability

Crystallographic structure factors and atomic co-ordinates have been deposited in the Protein Data Bank (PDB) under accession numbers 7Z8Y, 8B46 and 8B5X, and corresponding raw diffraction images have been deposited at https://proteindiffraction.org/. SAXS source data are included in Supplementary Data 1. All other data are available from the corresponding author on reasonable request.

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

## Acknowledgements
We thank Diamond Light Source and the staff of beamlines I24 and B21 (proposal MX25233), and M. Singleton for assistance with cryo-EM data collection. This work was supported by a Wellcome Senior Research Fellowship to O.R.D. (Grant Number 219413/Z/19/Z), and a core grant to the Wellcome Centre for Cell Biology (203149).

## Author contributions
M.G. and B.S.E. crystallised SUN1-KASH6 and performed biophysical experiments. O.R.D. solved the SUN1-KASH6 crystal structures, analysed data, designed experiments and wrote the manuscript.

## Competing interests
The authors declare no competing interest.
