## [Peer Review File · Communications Biology]

Reviewers' comments:

Reviewer #1 (Remarks to the Author):

The manuscript by Gurusaran M et al. reports the crystal structures of SUN1-KASH6. The structures show that the SUN-KASH6 forms heterocomplexes with a 9:6 stoichiometry, and this oligomeric state was demonstrated by biochemical experiments including SEC-MALS and solution SAXS. The authors argue that this unusual configuration is compatible with its biological function, as the complexes are positioned for the insertion of upstream sequences into the nuclear outer membrane.

Although the authors attempted to provide biochemical evidence for the organization of SUN-KASH6 in a 9:6 stoichiometry, it is still unable to exclude the possibility that such observations are due to a crystallographic artifact. This is because the formation of oligomeric states might be dynamic, depending on pH variations, and the result of crystallography symmetry (not in the asymmetric unit). They did not demonstrate the roles of protonation of the residues involved in the assembly. Furthermore, the biochemical experiments presented here used high concentrations of proteins.

Major comments:

In Figure 4, while the structures effectively illustrate the residues involved in the trimer-of-trimers assembly, it would be necessary to confirm whether these residues indeed facilitate the assembly process. It is necessary to provide evidence to demonstrate whether these residues are indeed important for interactions and oligomer assembly through in vitro assays such as pull-down assays or isothermal titration calorimetry (ITC) using mutants. These assays would help demonstrate the physical interactions or binding affinities, providing further support for the proposed assembly mechanism.

In Figure 5, when comparing SAXS data and the crystal structure, it has been convincingly demonstrated through the χ^2 value that the state of SUN1-KASH6 in solution varies depending on the pH, either existing in a 9:6 or 6:6 ratio. However, it is necessary to generate SAXS envelopes and align them with the crystal structures to provide more comprehensive insights.

The authors conducted SEC-MALS and solution SAXS experiments using a high concentration of proteins. To exclude the possibility of artifacts due to the high protein concentration, the authors should additionally include data using a reasonable concentration of protein.

The authors suggest that the LINC complex forms a 9:6 conformation under acidic conditions of pH 5.0. However, no evidence or reference information regarding the nucleus having a pH of 5.0 could be found in the manuscript. Please provide evidence for setting the experimental conditions at pH 5.0, including any additional information or references.

Minor review points:

It would be helpful for better understanding of the paper to have a diagram illustrating the protein domains and the crystallization construct before showcasing the crystal structure. This would provide valuable context regarding the protein's organization and aid in comprehending the subsequent analysis and findings.

In Figure 1a-c, the meaning of the boxes of various colors displayed on the body of the SUN trimer is not specified in the figure legend.

Reviewer #2 (Remarks to the Author):

The LINC complex spans across the inner and outer nuclear membranes of the nuclear envelope and transmits cytoskeletal forces (generated by molecular motors) into the nucleus to control the structure and movement of nuclear contents. The LINC complex is formed by SUN and KASH proteins that interact within the lumen of the nuclear envelope. A series of the structures of the SUN-KASH complex (formed by different types of SUN and KASH proteins) have revealed the plausible mechanism for assembling the LINC complex by SUN and KASH proteins.

In this manuscript, Gurusaran M., et al. reported the crystal structure of the SUN-KASH complex (formed between SUN1 and KASH6) in an asymmetric 9:6 conformation. The structure revealed that the KASH peptides in the complex adopt distinct conformations and are arranged with all six peptides emerging on the same molecular surface, which seems to be compatible with the insertion of the region preceding the KASH peptide into the outer nuclear membrane. They further demonstrated that the 9:6 complex can be formed in solution upon transient treatment of the 6:6 complex with the mildly acidic condition.

The asymmetric 9:6 conformation of the SUN1-KASH6 complex is of great interest and may provide a better explanation for organizing the LINC complex within the lumen of the nuclear envelope. However, more biochemical work and structural analysis are needed to substantiate their conclusion before the further consideration of this work to be published in *Communications Biology*.

1. Since the major finding of this work is the asymmetric 9:6 conformation, the authors should exclude the potential structural artifacts caused by crystal packing. Although the authors demonstrated the 9:6 state of the SUN1-KASH6 complex in solution, based on the structural analysis of the inter-domain interfaces, they need to further make a series of point mutations in these interfaces to evaluate their essential roles for assembling the 9:6 conformation. The mutational studies of the SUN1-KASH6 complex in solution would strengthen the conclusion and exclude the potential artifacts caused by crystal packing for the asymmetric 9:6 conformation.
2. The asymmetric 9:6 conformation of the SUN1-KASH6 complex is unexpected and distinct from the 6:6 conformation of other SUN-KASH complexes. The authors need to perform the extensive structural comparison of these two different conformations and summarize the key sites or inter-domain interfaces for the potential switch from the 6:6 to 9:6 conformation.
3. In Figure 1d, KASH5 is more divergent in terms of the KASH domain, while KASH6 is more divergent in terms of the transmembrane helix. The authors need to revise their statements about the divergence of KASH proteins in the main text.
4. It is interesting to note that, in the 9:6 complex, the third KASH-binding site in each SUN trimer seems to lose the KASH-binding capacity, which may be caused by the re-organization of the 6:6 complex (including two SUN trimers) to add one more SUN trimer to the complex. The authors need to include more discussion about this point and provide the potential explanation about the advantages of sacrificing the KASH-binding site of the SUN domain to re-arrange this 9:6 conformation.
5. Throughout the manuscript, the authors state that the 9:6 complex is directly compatible with the steric requirements of membrane insertion and the LINC complex assembly *in vivo*. However, it is unclear about the exact binding ratio between SUN and KASH proteins *in vivo* and the potential effect of the membrane on the LINC complex assembly. The reviewer suggest that the authors may need to

soften these statements unless they provide some direct evidence.

6. In Figure 5a-b, the authors observed that the SUN1-KASH6 complex undergoes the transition from the 6:6 to 9:6 conformation upon treatment with the mildly acidic condition and the N-terminal end of the KASH6 peptide seems to be capable of regulating this transition. Is it possible that the asymmetric 9:6 conformation is caused by the different truncations of the KASH peptide for crystallization? Since the N-terminal end of the KASH6 peptide is likely to be essential for the structural transition, the authors need to include more discussion and details about this segment and provide the potential explanation about its essential role.

7. Finally, regarding the mildly acidic condition for the structural transition, the authors need to include more discussion about the physiological condition (such as pH) within the lumen of the nuclear envelope to correlate the major finding of this work to its potential biological significance.

Reviewer #3 (Remarks to the Author):

The manuscript of O.R. Davies and co-workers describes the resolution of a new SUN-KASH crystal structure, between human SUN1 and JAW1/LRMP (KASH6). The group of O.R. Davies, as well as several other teams, have already published structural analyses of SUN-KASH complexes. These complexes play essential roles in nuclear organization. They form functionally critical physical links between the nucleoskeleton and the cytoskeleton. However, the new 3D structure described in this manuscript supports a different view of the SUN-KASH interface. It suggests a totally new model for this interface, which is not yet validated by cell biology studies but is in its-self very inspiring for designing new cell biology experiments. Indeed, the authors observed an unexpected stoichiometry (9:6 or 6:6) for their complex in the crystal state, which is more consistent than the previously observed SUN-KASH stoichiometry (6:6) with what is known from the position of the complex in the perinuclear space. They identified at very high resolution (between 1.7 and 2.3 Å) the substructures contributing to the interfaces between SUN molecules, KASH molecules, and SUN/KASH molecules. They found that such unexpected stoichiometry can also be observed in solution after incubation of the complex in specific acidic conditions. In general, the paper is well-written, the figures and their description are clear which is crucial given that the structural features to be described are relatively complicated. The structural work is robust, and the conclusions are well-supported by the experimental data. I have only minor remarks for the authors.

Introduction:

- "Their upstream sequences" -> do you mean "the SUN proteins"?
- "it was found that non-canonical KASH domains from Nesprin-4 and KASH5 form 6:6 SUN-KASH complexes" -> SUN1 or SUN2 or both?
- "Here, we report the crystal structure of the SUN-KASH complex between SUN1 and the DIVERGENT KASH domain of JAW1/LRMP" -> what does DIVERGENT mean here? This motif still shares some homology with other KASH motifs, as shown in Fig. 1a. I did not find in the introduction that the KASH domain of JAW1/LRMP is KASH 6, but that's how it is annotated in Fig. 1a, correct?

Results:

- Similarly I don't understand why in the second paragraph of the results, it is written " The recently identified KASH domain of JAW1/LRMP (KASH6) DIVERGES from other KASH proteins". It doesn't seem so clear to me that KASH6 shares less identity with the other KASH motifs.
- "Hence, SUN1-KASH6 underwent irreversible conformational change" -> what tells you that it is irreversible? What happens if you perform the SEC experiment at pH 5?

- "by size-exclusion chromatography small angle X-ray scattering" -> do the authors mean "size-exclusion chromatography coupled with small angle X-ray scattering"?

Discussion:

- It would be interesting to discuss which residues might be responsible for the conformational change at pH 5.
- It would also be interesting to design mutations that impair formation of the 9:6 complex and test the impact of these mutations on nuclear shape and Golgi structure; however, these experiments are clearly out of the scope of this article.

Response to reviewers' comments
'Crystal structure of SUN1-KASH6 reveals an asymmetric LINC complex architecture compatible with nuclear membrane insertion'
(COMMSBIO-23-2086-T)

We are grateful for the comments and suggestions of the reviewers. Here, we provide responses to the questions and comments raised, and outline how, in accordance with their requests, we have revised the manuscript (our revised text is highlighted in the revised manuscript).

Reviewer #1:

The manuscript by Gurusaran M et al. reports the crystal structures of SUN1-KASH6. The structures show that the SUN-KASH6 forms heterocomplexes with a 9:6 stoichiometry, and this oligomeric state was demonstrated by biochemical experiments including SEC-MALS and solution SAXS. The authors argue that this unusual configuration is compatible with its biological function, as the complexes are positioned for the insertion of upstream sequences into the nuclear outer membrane.

Although the authors attempted to provide biochemical evidence for the organization of SUN-KASH6 in a 9:6 stoichiometry, it is still unable to exclude the possibility that such observations are due to a crystallographic artifact. This is because the formation of oligomeric states might be dynamic, depending on pH variations, and the result of crystallography symmetry (not in the asymmetric unit). They did not demonstrate the roles of protonation of the residues involved in the assembly. Furthermore, the biochemical experiments presented here used high concentrations of proteins.

1. We agree that it is always imperative to ascertain the true oligomeric state of a protein in solution as it can be difficult to distinguish between oligomeric interfaces and crystal lattice contacts. In our original submission, we included SEC-MALS and SEC-SAXS data that confirm the formation of the 9:6 stoichiometry in solution. In response to the reviewer's comments, we are happy to provide additional data that further corroborate the 9:6 structure. These include SEC-MALS data at low protein concentrations, SEC-SAXS *ab initio* envelopes, structure-directed mutations that stabilise the structure in either 6:6 or 9:6 conformations, and cryo-EM data in which the 9:6 conformation is clearly observed in multiple reference-independent 2D class averages. We have described these additional data in our responses to the reviewers' comments below.

Major comments:

*In Figure 4, while the structures effectively illustrate the residues involved in the trimer-of-trimers assembly, it would be necessary to confirm whether these residues indeed facilitate the assembly process. It is necessary to provide evidence to demonstrate whether these residues are indeed important for interactions and oligomer assembly through *in vitro* assays such as pull-down assays or isothermal titration calorimetry (ITC) using mutants. These assays would*

help demonstrate the physical interactions or binding affinities, providing further support for the proposed assembly mechanism.

2. We agree that it is important to test whether these interactions are required for assembly. It is not possible to measure binding affinities by methods such as ITC as the complex must be co-expressed – SUN1 adopts an autoinhibited state in which KASH-binding cannot be achieved if expressed alone, as shown in the following papers (and corroborated by our observations):

Nie et al, 2016 (<https://doi.org/10.1016/j.str.2015.10.024>)

Xue et al, 2018 (<https://doi.org/10.1016/j.bbrc.2018.02.015>)

Jahed et al, 2018 (<https://doi.org/10.1091/mbc.E18-04-0266>)

Jahed et al, 2018 (<https://doi.org/10.1016/j.bpj.2018.01.015>)

As an alternative approach, we have introduced structure-directed mutations of residues at the trimer-of-trimers interface in co-purified complexes, and assessed their oligomeric states by SEC-MALS. Through this approach, we identified that two amino-acids – W534 and H539 – of the N-terminal α -helix of KASH6 α are required for 9:6 assembly. Specifically, separate W534A and H539Q mutations largely blocked 9:6 complex formation following treatment in acidic conditions. In contrast, an H539K mutation triggered the formation of 9:6 complexes without the need for treatment at low pH. Hence, our findings confirm that the interactions of W534 with KASH-lid β of an opposing trimer (shown in Figure 4a and new Figure 6a) mediates 9:6 complex formation. Our data further suggest that protonation of H539 is involved in the formation of the 9:6 complex following treatment at low pH as the lysine acts as a partial mimic in which the residue is positively charged at neutral pH, whilst the glutamine cannot be protonated. The protonated histidine likely forms a salt bridge with a backbone carbonyl of KASH-lid γ of the same trimer (shown in Figure 4b and new Figure 6b) to lock the N-terminal α -helix of KASH6 α in the conformation necessary for 9:6 complex formation. Further, we identified that H549 is also involved in 9:6 assembly. This amino-acid is located at the junction between the KASH peptide C-terminal insertion and β -interaction with the KASH-lid, and is a tyrosine in all other KASH sequences. The mutation H549Y partially blocked 9:6 assembly, with a large proportion of 6:6 complex remaining upon acidic treatment. Hence, protonation of H539 and H549, and interactions of W534 define the structural determinants of trimer-of-trimers assembly by SUN1-KASH6.

We have included the structure-directed mutagenesis data in a new Figure 6, in which we have provided additional structure panels that focus on the mutated amino-acids.

In Figure 5, when comparing SAXS data and the crystal structure, it has been convincingly demonstrated through the χ^2 value that the state of SUN1-KASH6 in solution varies depending on the pH, either existing in a 9:6 or 6:6 ratio. However, it is necessary to generate SAXS envelopes and align them with the crystal structures to provide more comprehensive insights.

3. We chose to show only the fits and χ^2 values as these provide the most sensitive tests of whether a structural model explains the solution SAXS data. The stark difference in fits between 9:6 and 6:6 models for treated/untreated samples strongly confirms our conclusions that the structure transitions from the 'classic' 6:6 structure to a 9:6 assembly following treatment at low pH. We are happy to provide the docked ab initio envelopes that the reviewer requests. These envelopes nicely predict the trimer-of-trimers and head-to-head nature of the treated/untreated molecules, and are closely fitted by docked models of the 9:6 and 6:6 complexes, respectively.

We have added the docked ab initio envelopes to Figure 5d.

Please note that we have also included new cryo-EM data in which the 9:6 trimer-of-trimers assembly is clearly visualised in reference-free 2D class averages (Figure 5e). This confirms our findings through an orthogonal approach.

The authors conducted SEC-MALS and solution SAXS experiments using a high concentration of proteins. To exclude the possibility of artifacts due to the high protein concentration, the authors should additionally include data using a reasonable concentration of protein.

4. We performed SEC-MALS and SEC-SAXS experiments with proteins loaded at 8-13 mg/ml. These concentrations were used to maximise the signal-noise ratio and thereby obtain the highest quality biophysical data. It is important to recognise that SEC has a dilution factor of over 5-fold, so the MALS and SAXS analyses are being performed at less than 2.5 mg/ml (approximately 10 μ M). Nevertheless, we have provided additional data (outlined below) that address these concerns.

We have included additional SEC-MALS data using a protein concentration of 0.1 mg/ml, corresponding to analysis following 5-fold dilution at less than 0.02 mg/ml (approximately 100 nM). This is the lowest concentration in which scattering data are sufficient for molecular weight determination (albeit with substantial noise). These data confirm formation of the 9:6 complex upon acidic treatment, indicating that trimer-of-trimers assembly occurs at high affinity (better than 100 nM apparent affinity). These data are included in Supplementary Figure 4.

It is not possible to perform SEC-SAXS at similarly low concentrations as the reduction in signal quality would provide such noisy data that it would be impossible to provide any meaningful structural insights. Indeed, with high noise, the χ^2 statistic would fail to discriminate between models that do or do not truly fit the data. Instead, we have used an orthogonal approach of cryo-EM to directly visualise the protein complex. This revealed reference-free 2D class averages in which the 9:6 trimer-of-trimers structure

was clearly visible. Hence, cryo-EM corroborates the formation of the 9:6 complex at lower protein concentrations. We have included these cryo-EM data in Figure 5e.

The authors suggest that the LINC complex forms a 9:6 conformation under acidic conditions of pH 5.0. However, no evidence or reference information regarding the nucleus having a pH of 5.0 could be found in the manuscript. Please provide evidence for setting the experimental conditions at pH 5.0, including any additional information or references.

5. We observed formation of the 9:6 assembly when the complex was exposed to pH 5.0 and then analysed in physiological conditions at pH 7.5 (Figure 5). In the new mutation data provided herein (described in response number 2), we have identified that this conformational change involves protonation of residue H539. As this change appears to be irreversible (i.e. it subsequently remains 9:6 at pH 7.5), it appears that acidic treatment overcomes an energy barrier that allows transition from the conformation adopted during expression in bacteria (6:6) to the stable conformation observed in the crystal structure (9:6). We envisage several possibilities for how this may occur in vivo. Firstly, the presence of chaperones in mammalian cells that are absent in bacteria may allow formation of the 9:6 rather than 6:6 complex. Secondly, an interacting protein may protonate the H539 residue. Thirdly, and in line with the reviewer's comments, a local pH in the range of 5.0 could induce protonation and stabilise the 9:6 complex.

In terms of the latter possibility, data regarding the pH within the mammalian nuclear envelope luminal space are rather lacking. In animal cells, the ER has a pH of around 7.2 (Paroutis et al, 2004, <https://doi.org/10.1152/physiol.00005.2004>), so it is likely that the luminal space is similar. Nevertheless, as the negative charge of lipid membranes attracts a proton cloud, biological membranes typically have an interface pH that is substantially lower than the bulk pH of the compartment. Indeed, a recent study of a membrane system determined an interface pH of between 3.9-5.3 (depending on lipid content) and a bulk pH of 6.8 (Parui et al, 2019, <https://doi.org/10.1039/C9SC02993A>). Hence, irrespective of the pH within the nuclear lumen, the close proximity of the SUN-KASH complex to the outer nuclear membrane may provide a local interface pH in an appropriate range to trigger assembly.

It is not possible for us to determine whether the 9:6 complex may form in vivo owing to folding chaperones, active protonation or local pH (or a combination of factors). Hence, in our original manuscript, we chose to present acidic treatment as a means of triggering a conformational change in vitro. Nevertheless, in light of the reviewer's comments, we have extended the discussion in the manuscript, including the above points to speculate on the possible means by which mammalian cells may assemble the 9:6 complex in vivo.

Minor review points:

It would be helpful for better understanding of the paper to have a diagram illustrating the protein domains and the crystallization construct before showcasing the crystal structure. This would provide valuable context regarding the protein's organization and aid in comprehending the subsequent analysis and findings.

6. We have added a schematic diagram to previous Figure 1d and have moved this to Figure 1a as it is sensible for it to be presented before models of the assembly.

In Figure 1a-c, the meaning of the boxes of various colors displayed on the body of the SUN trimer is not specified in the figure legend.

7. We have added a description to the legend, as requested.

Reviewer #2:

The LINC complex spans across the inner and outer nuclear membranes of the nuclear envelope and transmits cytoskeletal forces (generated by molecular motors) into the nucleus to control the structure and movement of nuclear contents. The LINC complex is formed by SUN and KASH proteins that interact within the lumen of the nuclear envelope. A series of the structures of the SUN-KASH complex (formed by different types of SUN and KASH proteins) have revealed the plausible mechanism for assembling the LINC complex by SUN and KASH proteins.

In this manuscript, Gurusaran M., et al. reported the crystal structure of the SUN-KASH complex (formed between SUN1 and KASH6) in an asymmetric 9:6 conformation. The structure revealed that the KASH peptides in the complex adopt distinct conformations and are arranged with all six peptides emerging on the same molecular surface, which seems to be compatible with the insertion of the region preceding the KASH peptide into the outer nuclear membrane. They further demonstrated that the 9:6 complex can be formed in solution upon transient treatment of the 6:6 complex with the mildly acidic condition.

The asymmetric 9:6 conformation of the SUN1-KASH6 complex is of great interest and may provide a better explanation for organizing the LINC complex within the lumen of the nuclear envelope. However, more biochemical work and structural analysis are needed to substantiate their conclusion before the further consideration of this work to be published in Communications Biology.

1. Since the major finding of this work is the asymmetric 9:6 conformation, the authors should exclude the potential structural artifacts caused by crystal packing. Although the authors demonstrated the 9:6 state of the SUN1-KASH6 complex in solution, based on the structural analysis of the inter-domain interfaces, they need to further make a series of point mutations in these interfaces to evaluate their essential roles for assembling the 9:6 conformation. The mutational studies of the SUN1-KASH6 complex in solution would strengthen the conclusion and exclude the potential artifacts caused by crystal packing for the asymmetric 9:6 conformation.

8. We strongly agree with the importance of confirming that the structure/oligomer observed in the crystal is the same as that in solution. In this case, we performed SEC-MALS to confirm that the 9:6 oligomer is formed in solution, and SEC-SAXS to confirm that the shape of the molecular in solution matches that of the 9:6 crystal structure. Hence, these data demonstrate that the 9:6 complex is not a consequence of crystal packing. We agree with the reviewer's suggestion of introducing point mutations at inter-domain interfaces. We have included these, and additional data supporting the 9:6 structure below.

We have introduced structure-directed mutations of residues at the trimer-of-trimers interface in co-purified complexes, and assessed their oligomeric states by SEC-MALS. Through this approach, we identified that two amino-acids – W534 and H539 – of the N-terminal α -helix of KASH6 α are required for 9:6 assembly. Specifically, separate

W534A and H539Q mutations largely blocked 9:6 complex formation following treatment in acidic conditions. In contrast, an H539K mutation triggered the formation of 9:6 complexes without the need for treatment at low pH. Hence, our findings confirm that the interactions of W534 with KASH-lid β of an opposing trimer (shown in Figure 4a and new Figure 6a) mediates 9:6 complex formation. Our data further suggest that protonation of H539 is involved in the formation of the 9:6 complex following treatment at low pH as the lysine acts as a partial mimic in which the residue is positively charged at neutral pH, whilst the glutamine cannot be protonated. The protonated histidine likely forms a salt bridge with a backbone carbonyl of KASH-lid γ of the same trimer (shown in Figure 4b and new Figure 6b) to lock the N-terminal α -helix of KASH6 α in the conformation necessary for 9:6 complex formation. Further, we identified that H549 is also involved in 9:6 assembly. This amino-acid is located at the junction between the KASH peptide C-terminal insertion and β -interaction with the KASH-lid, and is a tyrosine in all other KASH sequences. The mutation H549Y partially blocked 9:6 assembly, with a large proportion of 6:6 complex remaining upon acidic treatment. Hence, protonation of H539 and H549, and interactions of W534 define the structural determinants of trimer-of-trimers assembly by SUN1-KASH6.

We have included the structure-directed mutagenesis data in a new Figure 6, in which we have provided additional structure panels that focus on the mutated amino-acids.

In addition, we have used an orthogonal approach of cryo-EM to directly visualise the protein complex. This revealed reference-free 2D class averages in which the 9:6 trimer-of-trimers structure was clearly visible. Hence, cryo-EM corroborates the formation of the 9:6 complex at lower protein concentrations. We have included these cryo-EM data in Figure 5e.

Overall, our SEC-MALS, SEC-SAXS, cryo-EM and point mutation data strongly support our interpretation that the crystal structure corresponds to the 9:6 complex observed in solution.

2. The asymmetric 9:6 conformation of the SUN1-KASH6 complex is unexpected and distinct from the 6:6 conformation of other SUN-KASH complexes. The authors need to perform the extensive structural comparison of these two different conformations and summarize the key sites or inter-domain interfaces for the potential switch from the 6:6 to 9:6 conformation.

9. We have analysed the 9:6 structure and the 6:6 structure of other complexes. The main difference is the presence of the N-terminal α -helix that enables the KASH6 α peptide to adopt the unusual conformation that triggers the conformational change. Another important difference is the presence of histidine residue H549 at a site that is a tyrosine in other KASH sequences. These observations are confirmed by mutagenesis of the N-terminal α -helix and H549, which are described above in response number 8.

We have added additional discussion of the structural comparison as background for the mutagenesis described above, which we have illustrated in new Figures 6a-c.

3. *In Figure 1d, KASH5 is more divergent in terms of the KASH domain, while KASH6 is more divergent in terms of the transmembrane helix. The authors need to revise their statements about the divergence of KASH proteins in the main text.*

10. We intended to highlight that the KASH domain of KASH6 is divergent from the canonical KASH domain of Nesprin-1-3 and the non-canonical KASH domains of Nesprin-4 and KASH5). We have modified the text to clarify this point.

4. *It is interesting to note that, in the 9:6 complex, the third KASH-binding site in each SUN trimer seems to lose the KASH-binding capacity, which may be caused by the re-organization of the 6:6 complex (including two SUN trimers) to add one more SUN trimer to the complex. The authors need to include more discussion about this point and provide the potential explanation about the advantages of sacrificing the KASH-binding site of the SUN domain to re-arrange this 9:6 conformation.*

11. Yes, the third KASH-binding site is essentially lost upon formation of the trimer-of-trimers 9:6 structure. The reason for this is that the new assembly induces a low angulation of KASH-lid₃ that sterically hinders the ability of the third peptide to interact with this KASH-lid. Hence, the only available bind interface is the minimal binding pocket for the KASH peptide C-terminus. This likely retains only low binding affinity, explaining why we observed either no peptide or a few residues of peptide in our crystal structures. The retained residues of peptide in some structures likely reflect an artefact of its assembly from 6:6 complexes (in which peptides must be lost) rather than reflecting the true biological assembly. The clear advantage of losing the third peptide is that there is no simple topology in which the third peptide could reach a membrane on the same side of the protein as the other peptides. Hence, conformational change that removes (or at least greatly diminishes the third binding site) creates a structure in which two peptides can easily reach the same biological membrane.

As requested by the reviewer, we have added additional discussion of the above points to the manuscript.

5. *Throughout the manuscript, the authors state that the 9:6 complex is directly compatible with the steric requirements of membrane insertion and the LINC complex assembly in vivo. However, it is unclear about the exact binding ratio between SUN and KASH proteins in vivo and the potential effect of the membrane on the LINC complex assembly. The reviewer suggest that the authors may need to soften these statements unless they provide some direct evidence.*

12. We agree that the binding ratio and effects of the membrane are unknown, and are essentially unknowable given current technology (until cryo-EM achieves sufficient capabilities to resolve individual molecules in vivo within the nuclear lumen). Our statement is based on the assumption that the SUN-KASH complex is immediately below the membrane (assumed to be a plane), so geometry dictates that KASH peptides must emerge on one side of the molecule. We have added additional clarifications and

caveats throughout the text to highlight that these are assumptions based on theoretical considerations.

6. In Figure 5a-b, the authors observed that the SUN1-KASH6 complex undergoes the transition from the 6:6 to 9:6 conformation upon treatment with the mildly acidic condition and the N-terminal end of the KASH6 peptide seems to be capable of regulating this transition. Is it possible that the asymmetric 9:6 conformation is caused by the different truncations of the KASH peptide for crystallization? Since the N-terminal end of the KASH6 peptide is likely to be essential for the structural transition, the authors need to include more discussion and details about this segment and provide the potential explanation about its essential role.

13. We analysed two constructs of the SUN1-KASH6 construct, corresponding to KASH6 amino-acids 515-555 and 531-555. The former includes the full KASH domain and extends into the transmembrane region, whereas the latter is restricted to the amino-acids observed in the crystal structure. The transition from 6:6 to 9:6 structure was more complete for the latter than the former. However, we do not think that this relates to an ability of the N-terminal residues to regulate assembly. Instead, the unstructured N-terminal residues likely sterically interfere with assembly.

We have added a statement to the manuscript that enhancement of assembly is likely due to the steric effects of removing unstructured sequence.

7. Finally, regarding the mildly acidic condition for the structural transition, the authors need to included more discussion about the physiological condition (such as pH) within the lumen of the nuclear envelope to correlate the major finding of this work to its potential biological significance.

14. This is very similar to a comment provided by reviewer 1, which we have addressed in response number 5. For convenience, we have reproduced this response below.

We observed formation of the 9:6 assembly when the complex was exposed to pH 5.0 and then analysed in physiological conditions at pH 7.5 (Figure 5). In the new mutation data provided herein (described in response number 2), we have identified that this conformational change involves protonation of residue H539. As this change appears to be irreversible (i.e. it subsequently remains 9:6 at pH 7.5), it appears that acidic treatment overcomes an energy barrier that allows transition from the conformation adopted during expression in bacteria (6:6) to the stable conformation observed in the crystal structure (9:6). We envisage several possibilities for how this may occur in vivo. Firstly, the presence of chaperones in mammalian cells that are absent in bacteria may allow formation of the 9:6 rather than 6:6 complex. Secondly, an interacting protein may protonate the H539 residue. Thirdly, and in line with the reviewer's comments, a local pH in the range of 5.0 could induce protonation and stabilise the 9:6 complex.

In terms of the latter possibility, data regarding the pH within the mammalian nuclear envelope luminal space are rather lacking. In animal cells, the ER has a pH of around 7.2 (Paroutis et al, 2004, <https://doi.org/10.1152/physiol.00005.2004>), so it is likely that the luminal space is similar. Nevertheless, as the negative charge of lipid membranes attracts a proton cloud, biological membranes typically have an interface pH that is substantially lower than the bulk pH of the compartment. Indeed, a recent study of a membrane system determined an interface pH of between 3.9-5.3 (depending on lipid content) and a bulk pH of 6.8 (Parui et al, 2019, <https://doi.org/10.1039/C9SC02993A>). Hence, irrespective of the pH within the nuclear lumen, the close proximity of the SUN-KASH complex to the outer nuclear membrane may provide a local interface pH in an appropriate range to trigger assembly.

It is not possible for us to determine whether the 9:6 complex may form *in vivo* owing to folding chaperones, active protonation or local pH (or a combination of factors). Hence, in our original manuscript, we chose to present acidic treatment as a means of triggering a conformational change *in vitro*. Nevertheless, in light of the reviewer's comments, we have extended the discussion in the manuscript, including the above points to speculate on the possible means by which mammalian cells may assemble the 9:6 complex *in vivo*.

Reviewer #3:

The manuscript of O.R. Davies and co-workers describes the resolution of a new SUN-KASH crystal structure, between human SUN1 and JAW1/LRMP (KASH6). The group of O.R. Davies, as well as several other teams, have already published structural analyses of SUN-KASH complexes. These complexes play essential roles in nuclear organization. They form functionally critical physical links between the nucleoskeleton and the cytoskeleton. However, the new 3D structure described in this manuscript supports a different view of the SUN-KASH interface. It suggests a totally new model for this interface, which is not yet validated by cell biology studies but is in its-self very inspiring for designing new cell biology experiments. Indeed, the authors observed an unexpected stoichiometry (9:6 or 6:6) for their complex in the crystal state, which is more consistent than the previously observed SUN-KASH stoichiometry (6:6) with what is known from the position of the complex in the perinuclear space. They identified at very high resolution (between 1.7 and 2.3 Å) the substructures contributing to the interfaces between SUN molecules, KASH molecules, and SUN/KASH molecules. They found that such unexpected stoichiometry can also be observed in solution after incubation of the complex in specific acidic conditions. In general, the paper is well-written, the figures and their description are clear which is crucial given that the structural features to be described are relatively complicated. The structural work is robust, and the conclusions are well-supported by the experimental data. I have only minor remarks for the authors.

15. We greatly appreciate the positive comments of the reviewer and have addressed the minor comments, as outlined below.

Introduction:

- *“Their upstream sequences” -> do you mean “the SUN proteins”?*

16. We are referring the luminal regions of the SUN proteins (prior to SUN domains). We have modified the text to avoid any ambiguities.

- *“it was found that non-canonical KASH domains from Nesprin-4 and KASH5 form 6:6 SUN-KASH complexes” -> SUN1 or SUN2 or both?*

17. This applies both to SUN1 and SUN2. This is based on our crystal structures of SUN1 complexes and MALS analysis of SUN1 and SUN2 complexes (Gurusaran & Davies 2021, <https://doi.org/10.7554/eLife.60175>) and SUN2 crystal structures from others in which the 6:6 complexes are clear within the crystal symmetry (Cruz et al 2020, <https://doi.org/10.1016/j.jmb.2020.09.019>). We have clarified this point in the text.

- *“Here, we report the crystal structure of the SUN-KASH complex between SUN1 and the DIVERGENT KASH domain of JAW1/LRMP” -> what does DIVERGENT mean here? This motif still shares some homology with other KASH motifs, as shown in Fig. 1a. I did not find in the introduction that the KASH domain of JAW1/LRMP is KASH 6, but that’s how it is annotated in Fig. 1a, correct?*

18. By divergent, we mean that the KASH domain differs from the canonical KASH domain of Nesprin-1-3 (in a similar manner, we would say that KASH domains of Nesprin-4 and KASH5 are also divergent). This is more accurately referred to as being a non-canonical KASH domain. We have modified the text accordingly.

We initially defined the KASH domain of JAW1/LRMP as KASH6 at the start of the results section, but recognise that this is required in the introduction. Hence, we have modified the text to define KASH6 at an earlier stage of the manuscript.

Results:

- *Similarly I don't understand why in the second paragraph of the results, it is written "The recently identified KASH domain of JAW1/LRMP (KASH6) DIVERGES from other KASH proteins". It doesn't seem so clear to me that KASH6 shares less identity with the other KASH motifs.*

19. Here, we mean that its sequence diverges from the canonical KASH domains of Nesprin-1-3, and also from the non-canonical KAH domains of KASH4 and KASH5, so may adopt a distinct conformation. We have modified the text to clarify this point.

- *"Hence, SUN1-KASH6 underwent irreversible conformational change" -> what tells you that it is irreversible? What happens if you perform the SEC experiment at pH 5?*

20. By irreversible, we mean that the conformational change is retained following treatment at pH 5 when the experiment is performed at pH 7.5. This is in contrast to performing the biophysical experiments at pH 5, which we have done and show an even more complete formation of 9:6 complex. We have modified the text to clarify this point.

- *"by size-exclusion chromatography small angle X-ray scattering" -> do the authors mean "size-exclusion chromatography coupled with small angle X-ray scattering"?*

21. Yes. We have happily changed the text to use the reviewer's preferred terminology.

Discussion:

- *It would be interesting to discuss which residues might be responsible for the conformational change at pH 5.*

22. As part of our revisions, we have identified that residues W534, H539 and H549 participate in 9:6 complex formation, including through protonation of H539 (please see response number 2 for details). Hence, we have now included an additional Figure 6 in which we outline the molecular requirements responsible for conformational change at pH 5.

- It would also be interesting to design mutations that impair formation of the 9:6 complex and test the impact of these mutations on nuclear shape and Golgi structure; however, these experiments are clearly out of the scope of this article.

23. As outlined above, we have identified that residues W534, H539 and H549 participate in 9:6 complex formation, including through protonation of H539. Further, we have identified that 9:6 complex formation is blocked by separate mutations W534A and H539Q (and partially by H549Y), and is induced without an acidic trigger by mutation H539K (please see response number 2 for details). We agree that it is outside the scope of this manuscript to assess to perform mutagenesis in vivo. Nevertheless, by identifying and biochemically characterising these mutations, we will enable future work studying the biological roles of 6:6 and 9:6 complexes in vivo.

REVIEWERS' COMMENTS:

Reviewer #1 (Remarks to the Author):

In my opinion, the weak points were improved. I believe that the manuscript has now all the qualities to be published in Communications Biology.

Reviewer #2 (Remarks to the Author):

The authors have addressed all the reviewer's concerns. The manuscript has been significantly strengthened with the additional biochemical and structural data, and the reviewer supports the publication of this work in Communications Biology.

Reviewer #3 (Remarks to the Author):

The authors have carefully answered to all my questions.